
# Climate sensitivity of the summer runoff of two glacierised Himalayan catchments with contrasting climate

Sourav Laha[1,2], Argha Banerjee[1], Ajit Singh[2], Parmanand Sharma[2], and Meloth Thamban[2]

[1]Earth and Climate Science, Indian Institute of Science Education and Research (IISER) Pune, Pune-411008, India
[2]National Centre for Polar and Ocean Research (NCPOR), Ministry of Earth Sciences, Vasco-da-Gama, Goa-403804, India

**Correspondence:** Argha Banerjee (argha@iiserpune.ac.in)

**Abstract.** The future changes in runoff of Himalayan glacierised catchments will be determined by the local climate forcing and the climate sensitivity of the runoff. Here, we investigate the sensitivity of summer runoff to precipitation and temperature changes in winter-snow dominated Chandra (the western Himalaya) and summer-rain dominated upper Dudhkoshi (the eastern Himalaya) catchments. We analyse the interannual variability of summer runoff in these catchments during 1980–2018 using a
semi-distributed glacio-hydrological model, which is calibrated with the available runoff and glacier mass balance observations. Our results indicate that despite the contrasting precipitation regimes, the catchments have a similar runoff response: The summer runoff from the glacierised parts of both the catchments is sensitive to temperature changes and is insensitive to precipitation changes; the summer runoff from the non-glacierised parts has an exactly opposite pattern of sensitivity for both the catchments. The precipitation-independent glacier contribution stabilises the catchment runoff against precipitation
variability to some degree. The estimated sensitivities capture the characteristic 'peak water' in the long-term mean summer runoff, which is caused by the excess meltwater released by the shrinking ice reserve. As the glacier cover depletes, the summer runoff is expected to become more sensitive to precipitation forcing in these catchments. However, The net impact of the glacier loss on the catchment runoff may not be detectable, given the relatively large interannual runoff variability in these catchments.

## 1 Introduction

The presence of glaciers in a catchment significantly influences the diurnal to seasonal to interannual variability of the runoff, and its long-term multidecadal changes (Hock et al., 2005). Himalayan glacier-fed rivers play a key part in sustaining the downstream population and ecosystem (Azam et al., 2021). It is important to analyse the potential catchment-scale hydrological changes in the Himalaya as a significant reduction in the regional glacier cover by 2100 is expected (Kraaijenbrink et al., 2017).
This problem has motivated several glacio-hydrological model studies of the Himalayan basins and catchments (see Azam et al. (2021) for a review). These models often differ from each other in the level of descriptions of glacial processes, e.g., no explicit treatment of the glaciers (Pokhrel et al., 2014), a static (Nepal, 2016) or dynamic (Kraaijenbrink et al., 2017) glacier



cover, a simple temperature-index (Chandel and Ghosh, 2021; Banerjee, 2022) or detailed energy-balance based ice-melt model (Fujita and Sakai, 2014), and so on. Even a single model, when tuned with different available baseline climate data products, predicts a wide range of future hydrological changes (Koppes et al., 2015). In addition, the available future climate projections used to drive the glacio-hydrological models have a large spread (Sanjay et al., 2017). All of the above factors contribute to a wide range of predictions for the future changes in the runoff of Himalayan catchments (e.g. Nie et al., 2021). Assessing climate sensitivity of the runoff of Himalayan catchments may prove useful in reconciling the range of predictions available in the literature. Climate sensitivity of runoff is defined as the change in runoff due to a unit perturbation in a forcing variable, e.g., precipitation or temperature (Zheng et al., 2009). The climate sensitivities estimated from different models, which are forced by different projected climate forcing, can therefore be compared (Vano et al., 2012). A climate sensitivity analysis may also reveal key differences and similarities in the climate response of runoff generated from the different parts of a catchment (Banerjee, 2022) that are dominated by either snow melt or glacier melt or rainfall (Fujita and Sakai, 2014). It may also bring out the similarities and the differences among catchments across the Himalayan arc with their distinct climate settings, and thus, provide a better handle on the runoff response in the ungauged catchments in this data-sparse region (Azam et al., 2021).

In the literature, climate-sensitivities have been used to estimate long-term runoff changes due to temperature and precipitation forcing in both glacierised (Chen and Ohmura, 1990) and non-glacierised catchments (Dooge et al., 1999; Zheng et al., 2009; Vano et al., 2012). In the Himalaya, climate sensitivity of glacier mass balance proved useful to explain the observed spatial pattern of glacier thinning (Sakai and Fujita, 2017; Kumar et al., 2019), or to identify an inherent bias in scaling-based glacier evolution models which are often used in glacio-hydrological studies (Banerjee et al., 2020). Despite its potential utility, detailed studies of the climate sensitivity of the runoff of Himalayan glacierised catchments are limited (Fujita and Sakai, 2014; Azam and Srivastava, 2020; Banerjee, 2022). The present is motivated by this gap. A recent study used a simple temperature-index model to establish a weak precipitation sensitivity of glacier runoff in general (Banerjee, 2022). The present study uses a detailed process-based glacio-hydrological model to explore the sensitivity of glacier and off-glacier summer runoff, and the underlying mechanisms driving the sensitivities. We study two contrasting glacierised Himalayan catchments: winter-precipitation dominated Chandra (the western Himalaya), and summer-precipitation dominated upper Dudhkoshi (the eastern Himalaya). Climate sensitivities of runoff can be obtained simply by regressing the observed variability of runoff to those of its meteorological drivers (e.g. Zheng et al., 2009). When observations are not available, model simulations can be used for the same (Vano et al., 2012). Here, we use the Variable Infiltration Capacity (VIC) model (Liang et al., 1996) augmented with a glacier-melt module, to simulate runoff of the studied catchment over the period 1980–2018. The simulated runoff is used to estimate and validate the sensitivities of summer runoff to annual precipitation and summer temperature. The sensitivities of the runoff of the glacierised and non-glacierised parts of the catchments are also analysed separately. These sensitivities are then used to understand the multidecadal changes in the mean and the variability of summer runoff of the two catchments, as the glaciers shrink in a warming climate.

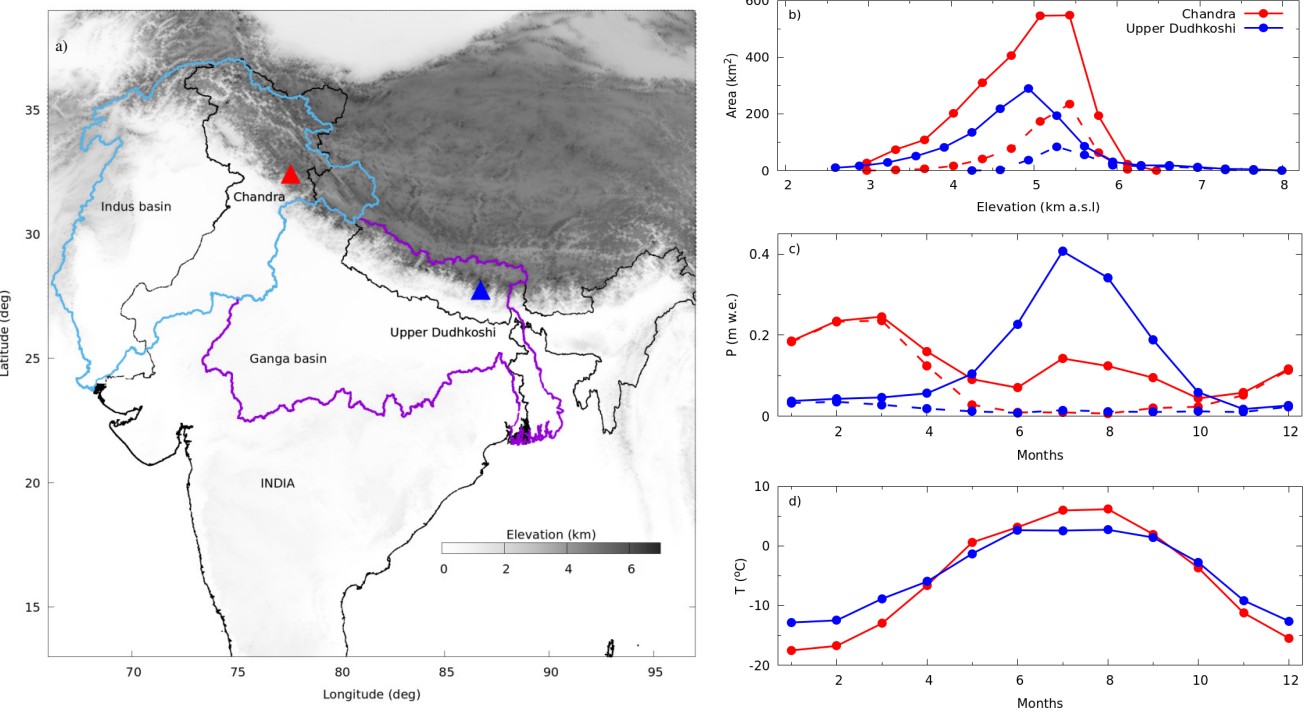

**Figure 1.** a) The location of Chandra (red solid triangle) and upper Dudhkoshi (blue solid triangle) catchments on a grey-scale elevation map (Amante et al., 2009). In the rest of the plots red (blue) colors refer to Chandra (upper Dudhkoshi) catchment. The solid magenta (sky-blue) polygon shows Ganga (Indus) basin. b) Area-elevation distribution of the catchments (solid lines + solid symbols), and that of the glacierised parts (dashed lines + solid symbols). c) Mean monthly precipitation (solid lines + solid symbols), along with the monthly snowfall (dashed lines + solid symbols). d) Mean monthly temperature profiles (solid lines + solid symbols).

## 2   Study area

We considered two high Himalayan catchments with contrasting climate regimes: Chandra (Indus basin, the western Himalaya), and upper Dudhkoshi (Ganga basin, the eastern Himalaya) (Figs. 1 and 2). Chandra catchment is in Lahaul-Spiti district, Himachal Pradesh, India. Upper Dudhkoshi catchment is located in Solukhumbu district of Nepal. About 70% of the annual precipitation in Chandra catchment occurs during the winter months (Fig. 1c) due to the Western Disturbances (Azam et al., 2019), and the influence of the Indian summer monsoon is relatively weak. In upper Dushkoshi catchment, more than 80% of the precipitation happens during the summer months (Fig. 1c) with a dominant influence of the Indian summer monsoon. Consequently, glacier accumulation mainly occurs during winter (summer) months in Chandra (upper Dudhkoshi) catchment. Due to the contrasting seasonality of precipitation, the ratio of liquid to solid precipitation in Chandra and upper Dudhkoshi catchments are 0.5 and 9.7, respectively. Despite the above differences, the two catchments are quite similar in terms of the





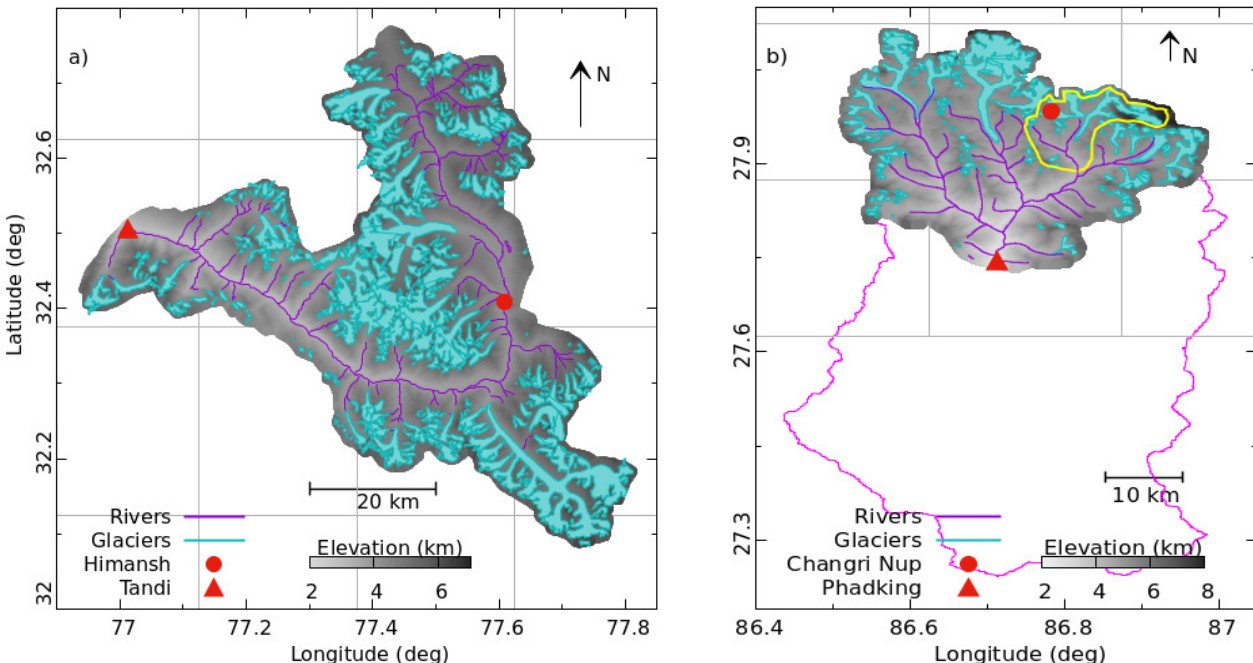

**Figure 2.** Maps of (a) Chandra and (b) upper Dudhkoshi catchments showing glaciers (Cyan polygons) and streams (purple lines). The red solid circles (triangles) are the meteorological (hydrological) stations. The ERA5 grid boxes are shown with solid gray lines in the background. Solid magenta and yellow polygons show Dudhkoshi and Periche catchments.

mean annual temperature, catchment hypsometry, elevation range, specific summer runoff, glacier fraction, and the recent rates of glacier loss (Table 1).

## 3   Data and methods

Below we present methodological details related to the input data, the glacio-hydrological model, and the climate sensitivity analysis. Note that throughout the paper, the annual quantities correspond to the hydrological year from 1st October of a calendar year to 30th September of the next, and summer season refers to the period from 1st May to 30th September (e.g., Azam et al., 2019).

### 3.1   Hydro-meteorological and glaciological data

#### 3.1.1   Observations

Observed hourly runoff of Chandra river at Tandi (32.55°N, 76.97°E, 2850 m a.s.l.) from 26th June, 2016 to 30th Oct, 2018 was available for three summer seasons with some data gaps (Fig. 5b) (Singh et al. (2020); supplementary Table S1). Hourly 2m





**Table 1.** A summary of the characteristics of Chandra and upper Dudhkoshi catchments. The meteorological variables are bias-corrected reanalysis data averaged over the catchments (Hersbach et al., 2020), the hydrological data are from model simulations (the present study). The glacier mass-balance and area-loss estimates are from the existing literature (supplementary Table S3).

| Catchment | Chandra | Upper Dudhkoshi |
| --- | --- | --- |
| Basin | Indus | Ganga |
| Area (km$^2$) | 2440 | 1190 |
| Outlet | Tandi | Phadking |
| Elevation range (m a.s.l) | 2850–6500 | 2600–7900 |
| Glacierised fraction | 0.25 | 0.20 |
| Annual temperature ($^\circ$C) | $-5.5$ | $-4.7$ |
| Annual precipitation (m yr$^{-1}$) | 1.6 | 1.5 |
| Summer precipitation / winter precipitation | 0.5 | 6.8 |
| Liquid precipitation/ solid precipitation | 0.5 | 9.7 |
| Glacier area loss (% decade$^{-1}$) | 1.1–5.5 | 1.2–4.2 |
| Glacier mass balance (m w.e. yr$^{-1}$) | $-0.13 \pm 0.11$ to $-0.56 \pm 0.38$ | $-0.26 \pm 0.13$ to $-0.52 \pm 0.22$ |
| Annual runoff (m yr$^{-1}$) | 1.25 | 0.99 |
| Summer runoff/annual runoff | 0.86 | 0.81 |

air temperature, precipitation, and incoming shortwave radiation were measured at the Himansh station (32.409$^\circ$N, 77.609$^\circ$E, 4080 m a.s.l.) in the catchment between 18th October, 2015 to 5th October, 2018 with some data gaps (Singh et al. (2020), supplementary Table S1).

Hourly runoff from upper Dudhkoshi catchment was observed at Phadking (27.74$^\circ$N, 86.71$^\circ$E, 2600 m a.s.l.) between 7th
April, 2010 and 16th April, 2017 (Fig. 5a) (Chevallier et al., 2017). Available hourly air temperature and precipitation data at Phadking from 7th April, 2010 to 23rd April, 2017 (with some data gaps) (Chevallier et al., 2017) were used. The daily incoming shortwave radiation data for the period 1st November, 2010 to 30th November, 2014 at nearby Changri Nup station (27.983$^\circ$N, 86.783$^\circ$E, 5400 m a.s.l.) in the same catchment were used (Sherpa et al. (2017); supplementary Table S1).

We considered eight available geodetic mass-balance observations that spanned a decade or more, for each of the catchments
(supplementary Table S3). Randolph Glacier Inventory (RGI 6) (Arendt et al., 2017) was used for the glacier boundaries that corresponded to the glacier extent in 2002.

### 3.1.2 Reanalysis data and bias correction

We used hourly 2-m air temperature, precipitation, and wind-speed from fifth-generation European Center for Medium-Range Weather Forecasts Atmospheric Reanalysis of the global climate (ERA5) from 1980 to 2018 (Hersbach et al., 2020) to force
the VIC model at a spatial resolution of 0.25$^\circ \times$0.25$^\circ$. Following the existing hydrological studies of various high Himalayan catchments (Soncini et al., 2016; Azam and Srivastava, 2020), the temperature data were bias-corrected. The available ob-





served air-temperature data at the Himansh station (Chandra catchment), and at Phadking (Dudhkoshi Catchment) were used
to compute the mean monthly temperature biases (supplementary Fig. S1), assumed to be constant for the whole catchment
and over the whole simulation period.

To compute temperature at any given elevation within a grid box, mean monthly lapse rates (supplementary Fig. S2) were
used. In Chandra catchment, the lapse rates were computed at the grid box containing Himansh station using ERA5 tempera-
ture from the four near-neighbour grid boxes. The corresponding annual lapse rate of $4.7\pm1.2$ °C km$^{-1}$ was consistent with
previously observed values of 4.4–6.4 °C km$^{-1}$ (Azam et al., 2019; Pratap et al., 2019). In upper Dudhkoshi catchment, the
monthly lapse rates derived from ERA5 were significantly larger than those observed between Phadking and Changri Nup
stations over the period 2013–2016, so we used the observed lapse rates. The corresponding mean annual lapse rate of $4.6\pm0.6$
°C km$^{-1}$ in this catchment was the same as that previously reported (Pokhrel et al., 2014).

    ERA5 precipitation data was corrected by scaling with a catchment-specific constant $\alpha_P$ for each of the catchments follow-
ing the existing studies from the region (Huss and Hock, 2015; Bhattacharya et al., 2019; Azam and Srivastava, 2020). The
scale factor, which ensured water balance over the catchments, was calibrated using the observed runoff and glacier mass-
balance employing a Bayesian procedure (see Sect. 3.2.4). In some of the existing studies in the region, an elevation-dependent
precipitation scaling has also been employed (e.g., Azam et al., 2019). However, as an elevation-dependent correction may
potentially introduce additional uncertainties (e.g., Johnson and Rupper, 2020), we preferred a constant $\alpha_p$ keeping the number
of calibration parameters to a minimum. Note that the precipitation biases over the rugged Himalayan catchments ($\sim$1000
km$^2$) cannot be corrected using data from a single station because of a high spatial variability and a small correlation length
associated with precipitation (Singh and Kumar, 1997).

    We scaled the incoming shortwave radiation obtained from VIC model by a catchment-specific constant to match the cor-
responding mean values observed at Himansh (Chandra catchment) and Changri Nup (upper Dudhkoshi catchment) stations
(supplementary Fig. S3).

## 3.2   Glacio-hydrological model setup

We divided each studied catchment into two parts, the glacierised and non-glacierised ones. On the non-glacierised part, we
ran another VIC model (Liang et al., 1996) to compute the surface runoff, baseflow, and evapotranspiration at hourly time
steps (Fig. 3). On the glacierised part, a VIC model was used to get the snow melt and a temperature-index model (Hock,
2003) to obtain the glacier melt (Fig. 3). The additional glacier module was needed as VIC model does not have the capability
to compute glacier melt (Liang et al., 1996). A similar approach to represent glacier melt was used in existing VIC model
studies in the region (Zhang et al., 2013; Zhao et al., 2015; Chandel and Ghosh, 2021). Hourly hydrological fluxes of the
non-glacierised and glacierised parts within each gridbox were combined and routed (Lohmann et al., 1998) to obtain the total
runoff at the catchment outlet. The flow from each gridbox was partitioned into the fast and slow components using hydrographs
parameterised with $Bf$ and $Ks$, $UH_{max}^F$, and $UH_{pow}^F$ (Lohmann et al., 1998). The total hourly runoff produced from each
grid box was routed downstream in the direction of steepest descent using a linearised Saint-Venant equation (Lohmann et al.,
125  1998).







**Figure 3.** Flow chart of the glacio-hydrological model setup (see Sect. 3.3 for details).





### 3.2.1 Hydrological model

VIC (version 4.2.d, accessible from https://vic.readthedocs.io/en/master/; Liang et al. (1996)) is a semi-distributed macro-scale hydrological model, which simulates the fluxes of water and energy for a grid-based representation of a catchment using physically-based parameterisations of hydrological processes (Liang et al., 1996). In this model, water can enter a gridbox only from the atmosphere, and once water reaches the river channel, it can not flow back into the gridbox. These assumptions limit the applicability of the model to a larger grid size (e.g., a grid size of $0.25°$ which was used here). The VIC model considers sub-grid heterogeneity in surface topography, land-cover, and sub-surface soil properties. Different vegetation classes are represented by tiles covering a fraction of the gridbox, and an area-weighted sum over the tiles obtains various hydrological fluxes for each gridbox. VIC model partitions the input precipitation at each gridbox into rain and snow based on a threshold temperature $T_{th}$. It uses a two-layered snowpack, computing the snow melt at a given elevation with an energy-balance approach. A surface-albedo parameterisation incorporating the effects of snowfall and aging of snow, snow-sublimation, and refreezing of meltwater within the snowpack are included in the model (Andreadis et al., 2009). Evapotranspiration is computed by Penman-Monteith equation (Liang et al., 1996) as the sum of canopy evaporation, bare soil evaporation, and transpiration for each vegetation class. VIC allows multiple subsurface soil layers, and here we used three of them. The partitioning between surface runoff and infiltration into the top layer is done using a variable infiltration curve (Liang et al., 1996) controlled by the parameter $b_{inf}$. The bottom layer produces the baseflow depending on the moisture content with a maximum allowed baseflow of $Ds_{max}$. At low soil moisture (below a fraction $Ws$ of the maximum allowed soil moisture, and up to a fraction $Ds$ of $Ds_{max}$), the baseflow is linear in it. Beyond this linear regime, a non-linear ARNO recession curve determines baseflow (Liang et al., 1996). The chosen values of the above five VIC model parameters are given in supplementary Table S2.

Dictated by the resolution of ERA5 input data, the model was run at a $0.25° × 0.25°$ spatial resolution and at an hourly time steps. Chandra (upper Dudhkoshi) catchment covered parts of 11 (6) ERA5 grid boxes (Figs. 2a–2b), with fractional grid cover in the range 2.5–92% (2–68%). The static input parameters included soil properties (Nachtergaele et al., 2010), land use (Friedl and Sulla, 2019), vegetation information (Rodell et al., 2004), and elevation distribution (Farr et al., 2007) for each gridbox. We used 10 elevation bands with width in the range 100–300 m depending on the elevation range within the gridbox. A minimal set of meteorological forcing parameters, namely, bias-corrected air temperature, scaled precipitation, and wind speed from ERA5 reanalysis (Hersbach et al., 2020) over the period 1980–2018 were used to force the model. For model spin-up, we extended the meteorological input data back by repeating the data from 1980 to 1984.

### 3.2.2 Glacier model

On the glacierised grids of each catchment, a separate VIC model computed the snow melt and snow-covered fraction of each elevation band (Fig. 3). A temperature-index model (Hock, 2003) obtained the ice melt over the corresponding snow-free areas. The glacier melt module was forced with the bias-corrected ERA5 air temperature, while taking into account the elevation of the band using a mean monthly lapse rate (supplementary Fig. S2). The degree-day factors (DDF) for each of the catchments were calibrated simultaneously against the observed glacier mass balance and catchment runoff using a Bayesian method (see



Sect. 3.3.4). The snow melt, ice melt, and rainfall on the glaciers routed through two linear reservoirs (Hannah and Gurnell,
2001) with reservoir constants $K_{fast}$ and $K_{slow}$, obtained the glacier runoff (e.g., Radić and Hock, 2014). Catchment-wise
glacier mass balance were computed by subtracting the total ice and snow melt from the total snowfall over the glacierised
parts. The present glacier module did not consider snow redistribution within or between the glacierised and the non-glacierised
parts of the catchment via avalanching (Laha et al., 2017) or wind redistribution, and the effect of supraglacial debris layer on
melting (Kraaijenbrink et al., 2017). We did not consider any baseflow contribution from the glacierised parts.

We assumed a static glacier cover here as the observed percentage loss of glacier area over the simulation period was small
(1–5 % decade$^{-1}$) for both the catchments (Table 1). Biases due to such a static-glacier approximation were found to be small
for another glacierised Himalayan catchment over the same period (Azam and Srivastava, 2020). A dynamic description of
glaciers within the glacio-hydrological model is needed only for predicting the long-term changes in runoff when potential
changes in glacier extent is large (e.g., Kraaijenbrink et al., 2017).

**3.2.3   Model calibration**

With the limited set of observations available for the studied catchments, calibrating a large number of tunable parameters
may not ensure a better representation of the relevant processes (Jost et al., 2012), and may lead to over-fitting. It may also
suffer from equifinality (Beven and Freer, 2001; Jost et al., 2012), where more than one parameter combinations reproduce
the observed runoff. These issues are likely to compromise the ability of the calibrated model to capture the responses of
the glacierised and the non-glacierised parts of the catchments. Therefore, here we calibrated only two model parameters: 1)
precipitation scale factor $\alpha_P$, and 2) DDF of ice, as they determine the catchment-wide water balance and glacier mass balance.
For the rest of the VIC model parameters, we used the central values of the recommended range (supplementary Table S2).
Note that, these uncalibrated VIC model parameters values were similar to that of the corresponding calibrated values used in
some of the studies from the region (e.g., Zhang et al., 2013; Zhao et al., 2015; Bhattacharya et al., 2019; Chandel and Ghosh,
2021). This suggested that the VIC model parameters used here to describe the two Himalayan catchments were representative
ones. These model parameter values are listed in supplementary Table S2.

To calibrate for the parameters $\alpha_P$ and DDF, we used a Bayesian method (e.g., Tarantola, 2005). For given a set of available
observations $d$ and a set of model parameters $\theta$, the posterior probability of the model parameters given the observations was,

$$p(\theta|d) \propto p(d|\theta)p(\theta). \tag{1}$$

Here $p(\theta)$ was the prior distribution of the model parameters $\alpha_P$ and DDF. We assumed a uniform prior distribution over the
range of values reported over the High Mountain Asia: 0.7–2.5 for $\alpha_P$ (Huss and Hock, 2015; Bhattacharya et al., 2019; Azam
and Srivastava, 2020), and 2–16 mm °C$^{-1}$ day$^{-1}$ for DDF (Singh et al., 2000; Nepal, 2016; Azam et al., 2019). The conditional
probability $p(d|\theta)$ of the observations $d$ given the model parameter $\theta$ was assumed to be,

$$p(d|\theta) \sim e^{-\frac{\sum_i (Q_i^{mod}-Q_i^{obs})^2}{2\sigma_Q}} \times e^{-\frac{1}{2}\frac{\sum_j (b_j^{mod}-b_j^{obs})^2}{2\sigma_b}}. \tag{2}$$





Here the superscript $obs$ and $mod$ denoted the observed and modelled values, respectively. The total summer runoff for the $i$-th

year was $Q_i$, and the summation was over all the years with observed runoff data. The uncertainty $\sigma_Q$ in summer runoff was

taken to be $\sim$10% of the mean summer runoff, which is a conservative estimate given the previously reported 5% error for other

Himalayan rivers (e.g., Singh et al., 2005). The $j$-th observed regional geodetic glacier mass balance for each catchment was

denoted by $b_j$. This summation was over eight such observations (Bolch et al., 2011; Gardelle et al., 2012; Nuimura et al., 2012;

Vincent et al., 2013; Vijay and Braun, 2016; Brun et al., 2017; King et al., 2017; Mukherjee et al., 2018; Maurer et al., 2019;

Shean et al., 2020) for each of the catchment as listed in the supplementary Table S3. The corresponding median uncertainties

was 0.13 m w.e yr$^{-1}$ (supplementary Table S3). An empirical factor of $\frac{1}{2}$ in the second exponent in Eq. (2) ensured that the

two exponential weights were of similar magnitude for the most-probable model. For each catchment, the two-dimensional

parameter space was scanned with step sizes of 0.2 for $\alpha_P$, and 0.5 mm $°C^{-1}$ day$^{-1}$ for DDF. This yielded an ensemble of

$11 \times 29 = 319$ models for each catchmen, with associated weight $p(\theta|d)$ as computed using Eq. (1).

### 3.2.4   Model validation, parameter sensitivity, and uncertainty

The results from the most-probable model were used for estimating summer runoff and its components, and glacier mass

balance. The weighted ensemble of all the 319 models was used to obtain the corresponding $2\sigma$ uncertainties.

To assess the model performance, the simulated mean summer runoff, decadal glacier mass balance, and glacier melt contri-

bution were compared with the corresponding modelled and observed values previously reported in the region. As the observed

runoff was available for only 3 to 7 years, all of it was utilised for the above calibration without any validation period. For

upper Dudhkoshi catchment the calibration procedure was repeated using data from a set of four consecutive years, while the

remaining three year's data were utilised for validation. This experiment was repeated four times with different choices of

calibration period.

Following earlier studies (e.g., He and Pang, 2015), the parameter sensitivity of the results of the most-probable model was

estimated with the help of additional 22 simulations where one of the 11 glacio-hydrological model parameters (supplementary

Table S2) was perturbed by $\pm$25% of the range of corresponding recommended values. The sensitivity of summer runoff to

these 11 parameters were computed at the corresponding optimal values of DDF and $\alpha_P$. Perturbing the parameters one by

one in the 11-d parameter space is similar to computing the multidimensional gradient in this space to understand the model

sensitivity. An ensemble of 200 model outputs was generated where one of the above 13 parameters of the best-fit model

perturbed by $\pm$25%. The mean and standard deviation of this ensemble were used to assess the parameter sensitivity. To look

for possible interactions between parameters, 80 additional simulations were ran, where a randomly chosen pair of parameters

were simultaneously perturbed.

### 3.3   Climate sensitivity of summer runoff

The climate sensitivity of specific summer runoff $Q$ (m yr$^{-1}$) is defined as the change in runoff due to a unit perturbation

in a meteorological forcing parameter (e.g., Zheng et al., 2009). Here, we considered the sensitivity of summer runoff $Q$ due

to changes in annual precipitation $P$ (m yr$^{-1}$) and mean summer temperature $T$ (°C), as summer runoff was $81 - 86$% of





the annual runoff in these catchments (Table 1). We did not consider the annual or winter temperature as it is the summer temperature that controls glacier melt (e.g., Pratap et al., 2019).

### 3.3.1 Climate sensitivities and summer runoff anomalies

The sensitivities of summer runoff relate (e.g., Zheng et al., 2009) the anomalies of summer runoff $\delta Q$ (m yr$^{-1}$), annual precipitation $\delta P$ (m), and summer air-temperature $\delta T$ (°C) as follows.

$$\delta Q = s_P \delta P + s_T \delta T. \tag{3}$$

Here, precipitation sensitivity is denoted by $s_P \doteq \frac{\partial Q}{\partial P} = \frac{\partial \delta Q}{\partial P}$ (m yr$^{-1}$ m$^{-1}$), and temperature sensitivity is denoted by $s_T \doteq \frac{\partial Q}{\partial T} = \frac{\partial \delta Q}{\partial T}$ (m yr$^{-1}$ °C$^{-1}$). In Eq. (3), a possible bilinear interaction term proportional to $\delta T \delta P$ (Lang, 1986) was not considered. We confirmed this correction term, when included in the regression for the catchment studied, was not significant ($p < 0.05$). In order to estimate the sensitivities $s_T$ and $s_P$ (Eq. (3)), we regressed simulated time series of $\delta Q$ for the catchments during 1997–2018 with the corresponding time series of $\delta T$ and $\delta P$. The standard error of the fits obtained the corresponding uncertainties. The sensitivities estimated from the simulated $\delta Q$ time series over 1997–2018 were validated using that during 1980–1996 by computing the corresponding Nash-Sutcliffe efficiency (NSE) and root mean squared error (RMSE).

We also considered the runoff from glacierised part of the catchments $Q^{(g)} \doteq Q_0^{(g)} + \delta Q^{(g)}$, and that from the non-glacierised part of the catchments $Q^{(r)} \doteq Q_0^{(r)} + \delta Q^{(r)}$. Here, the notations $Q_0$ and $\delta Q$ denote the long-term mean and the anomaly for a given year, respectively. The corresponding sensitivities were defined in a similar way and led to the relations,

$$\delta Q^{(g)} = s_P^{(g)} \delta P + s_T^{(g)} \delta T, \tag{4}$$

$$\delta Q^{(r)} = s_P^{(r)} \delta P + s_T^{(r)} \delta T. \tag{5}$$

The climate sensitivities of glacierised and non-glacierised parts (Eqs. (4) and (5)) and the corresponding uncertainties were estimated in the same way as above using the anomalies $\delta Q^{(g)}$ and $\delta Q^{(r)}$, along with $\delta P$ and $\delta T$.

Given the instantaneous glacier fraction $x$, the quantities defined for the glacierised and non-glacierised part of the catchments are related to those defined for the whole catchment as,

$$\delta Q = x \delta Q^{(g)} + (1-x) \delta Q^{(r)}, \tag{6}$$

$$s_T = x s_T^{(g)} + (1-x) s_T^{(r)}, \tag{7}$$

$$s_P = x s_P^{(g)} + (1-x) s_P^{(r)}. \tag{8}$$

Apart from the sensitivities of summer runoff, we also computed the precipitation and temperature sensitivities of glacier mass balance using the corresponding simulated interannual variability over the period of 1980–2018. The precipitation sensitivity of glacier mass balance was defined to be the mass-balance change due to a 10% change in precipitation following the convention used in the literature (e.g., Wang et al., 2019).





### 3.3.2 Variability of summer runoff

The climate sensitivities defined above allow determination of the interannual variability of summer runoff given those of $P$ and $T$,

$$\sigma_Q = \sqrt{s_T^2 \sigma_T^2 + s_P^2 \sigma_P^2}, \tag{9}$$

where $\sigma_Q, \sigma_P$, and $\sigma_T$ are standard deviations of $Q$, $P$, and $T$, respectively. An implicit assumption here is that $\delta P$ and $\delta T$ are uncorrelated over the simulation period, which we verified to be true at $p < 0.05$ level for both the catchments.

We computed $\sigma_P$ and $\sigma_T$ during 1980–1996 and 1997–2018 from the forcing data, and used Eq. (9) to predict the corresponding $\sigma_Q$. These predictions were validated using the corresponding $\sigma_Q$ obtained directly from the simulated summer runoff time series. We analysed the future changes in $\sigma_Q$ in the studied catchments due to shrinking glaciers, and the variation of $\sigma_Q$ for a set of hypothetical catchments having different $x$. Note that an empirical non-monotonic dependence of the coefficient of variation of runoff across catchments on the corresponding fractional glacier cover with a minimum at a moderate glacier cover has been termed as 'glacier-compensation effect' (Chen and Ohmura, 1990).

### 3.3.3 Long-term changes in mean summer runoff

The climate sensitivities defined above can be used to predict the multidecadal changes in summer runoff ($\Delta Q$) for given changes in annual precipitation ($\Delta P$) and mean summer temperature ($\Delta T$). For a change in glacier fraction $\Delta x$ from the initial value of $x_0$ (i.e., $x \doteq x_0 + \Delta x$), the following linear-response equation can be constructed ignoring the terms that were higher order in $\Delta$.

$$\begin{aligned}
\Delta Q &= x(s_P^{(g)} \Delta P + s_T^{(g)} \Delta T) + (1-x)(s_P^{(r)} \Delta P + s_T^{(r)} \Delta T) \\
&\quad + \Delta x(Q_0^{(g)} - Q_0^{(r)}).
\end{aligned} \tag{10}$$

A similar linear-response approach was used to analyse glacier-compensation effect (Chen and Ohmura, 1990) without explicitly referring to climate sensitivity. As ERA5 annual precipitation showed low/little spatial variability within the two catchments (supplementary Fig. S4), here we ignored the spatial variation of the generated runoff within the off-glacier or glacierised areas. We also assumed that the contribution of the deglacierised area to the changes in summer runoff is well represented by the difference between the mean runoff of the glacierised and non-glacierised parts. Note that climate-sensitivity based predictions for future changes in runoff are reliable as long as the predicted changes lie within the range of the recent interannual variability of $P, T$ and $Q$. Beyond this range, there may be uncontrolled extrapolation errors.

Equation (10) was used to investigate the multi-decadal changes in the summer runoff, assuming glacier-loss scenarios in Chandra and upper Dudhkoshi catchments to be the same as those projected for Indus and Ganga basin under RCP 2.6 climate scenario (Huss and Hock, 2018). The corresponding temperature projections were obtained from available estimates for the western and eastern Himalaya, respectively (Kraaijenbrink et al., 2017). The related precipitation changes, which were not significant within the uncertainties for both the regions (Kraaijenbrink et al., 2017), were ignored here. Consequently, the terms with $\Delta P$ in Eq. (10) did not contribute to the estimated changes.



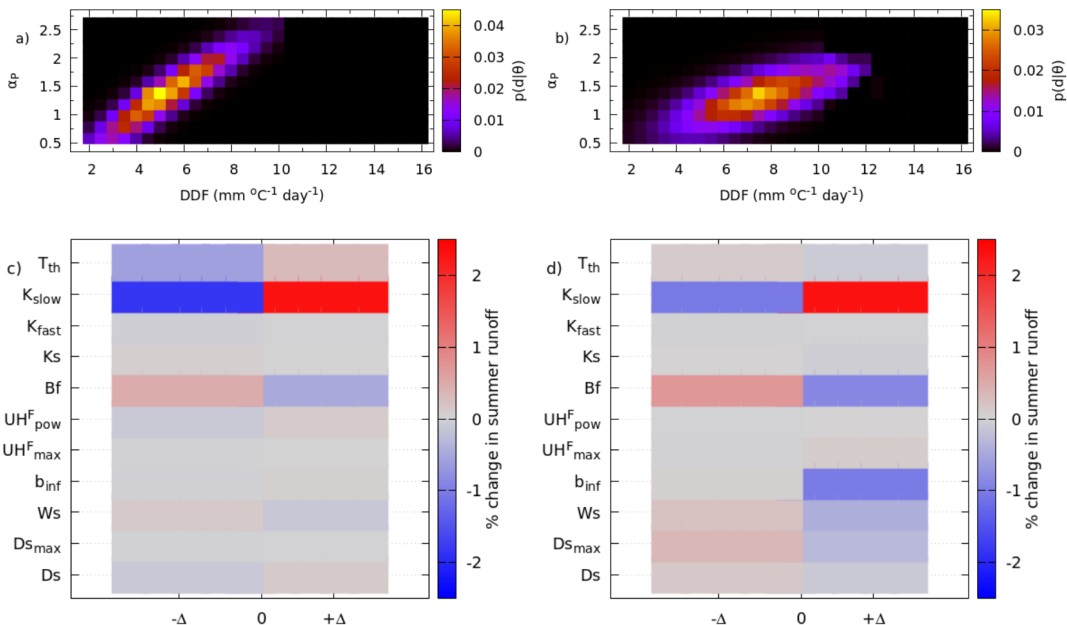

**Figure 4.** (a) and (b) shows the posterior probability distribution $p(d|\theta)$ of the model parameters ($\alpha_P$, DDF) for Chandra and upper Dushkoshi catchment, respectively (see Sect. 4.1). (c) and (d) shows the sensitivities of the simulated summer runoff to perturbations in 11 VIC the model parameters for Chandra and upper Dushkoshi catchments, respectively. Here, $\pm\Delta$ denotes the perturbation of parameters by $\pm 25\%$ of the corresponding prescribed range (see Sect. 3.3.1–3.3.2, and supplementary Table S2).

Under a sustained warming, glacier runoff is expected to show a peak over a multidecadal scale due to the excess meltwater

contribution from the shrinking glacier reserve, which is followed by a decline in the runoff as the ice reserve depletes (Huss and Hock, 2018). Following (Huss and Hock, 2018), we defined 'peak water' as the maximum change in runoff of the area that was glacierised at 2000 AD, and used Eq. (10) to predict the timing and the magnitude of the 'peak water' in the studied catchments. While the glacier boundaries (Arendt et al., 2017) belonged to 2002, the small changes in glacier area between 2000 and 2002 was ignored for this calculation due to an observed slow rate of glacier area change (Table 1).

**4   Results and discussions**

**4.1   Calibration and validation**

The Bayesian calibration method fitted the observed glacier mass balance and the summer runoff data simultaneously, that yield unique best-fit models for both the catchments (Figs. 4a–4b). Thus, the present calibration strategy resolved the equifinality problem that is usually encountered while calibrating glacio-hydrological models using only discharge data (e.g., Azam and

Srivastava, 2020). The most-probable DDF values were 5.0 and 7.5 mm day$^{-1}$ °C$^{-1}$ for Chandra and upper Dudhkoshi catchments, respectively. These DDF values were in the same ballpark range as previously used in studies in and around



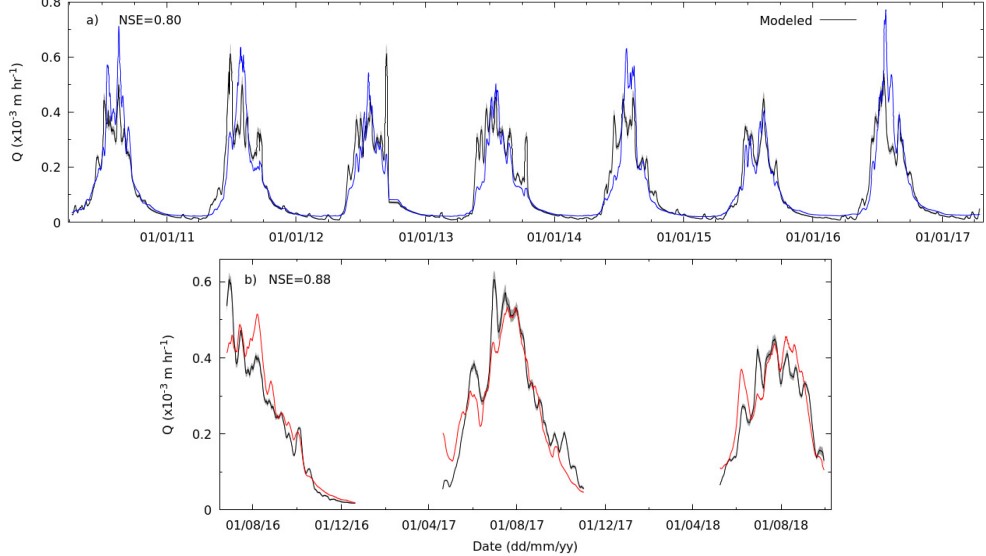

**Figure 5.** Modelled weekly runoff (black lines, with grey bands denoting 2-$\sigma$ uncertainty) compared with the corresponding observations for (a) upper Dudhkoshi (blue solid line), and (b) Chandra (red solid line) catchments.

Chandra (Azam et al., 2019; Pratap et al., 2019) and Dudhkoshi catchments (Pokhrel et al., 2014; Khadka et al., 2014; Nepal, 2016). The best-fit $\alpha_P$ was 1.4 for both the catchments which was within the range of values 0.7–1.5 used in the existing studies in the Himalaya to correct various reanalysis products (Huss and Hock, 2015; Bhattacharya et al., 2019; Azam and

Srivastava, 2020).

The calibrated models reproduced the observed summer runoff of the catchments reasonably well (Fig. 5) with RMSEs of 11 and 12% of the mean summer runoff, and NSEs of 0.88 and 0.80 for Chandra and upper Dudhkoshi catchments, respectively. These RMSE and NSE values were comparable to or smaller than those reported in the existing studies from the region (Nepal, 2016; Mimeau et al., 2018; Bhattacharya et al., 2019; Azam et al., 2019; Azam and Srivastava, 2020). Four additional

calibration experiments for upper Dudhkoshi catchment, each one using a different set of 4 consecutive years of runoff data for calibration, obtained most-probable models with DDF (7.2±1.1 mm day$^{-1}$ °C$^{-1}$), $\alpha_P$ (1.43 ± 0.03), NSEs (0.79–0.86), and RMSEs (10–14% of mean summer runoff) similar to those mentioned above.

## 4.2 Simulated runoff and its parameter sensitivity

The simulated mean summer runoff of Chandra and upper Dudhkoshi catchments over the period 1980–2018 were 1.08±0.03

and 0.81±0.02 m yr$^{-1}$, respectively (supplementary Fig. S7). The corresponding standard deviations were 0.14 and 0.10 m yr$^{-1}$. The mean summer runoff of the glacierised and the non-glacierised parts of Chandra catchment were 1.54 and 0.92 m yr$^{-1}$, respectively. The corresponding values for upper Dudhkoshi catchment were 1.59 and 0.61 m yr$^{-1}$. In these two catchments, more than 81% of the simulated annual runoff were during the summer season. In comparison, seven years of





observation from upper Dudhkoshi catchment (Chevallier et al., 2017) showed a mean specific summer runoff of $0.86\pm0.05$
m yr$^{-1}$, which was 83% of the mean annual runoff. Our simulations indicated that glacier runoff contributed $39\pm9\%$ and
$36\pm11\%$ of the total summer runoff in upper Dudhkoshi and Chandra catchments, with the glacier ice loss amounting to 9%
and 4% of the respective total summer runoff (supplementary Fig. S7).

Existing model studies reported annual runoff of 1.6 m yr$^{-1}$ during 2000–2010 (Nepal, 2016) and 0.96 m yr$^{-1}$ during
1981–2015 (Chandel and Ghosh, 2021) for the whole Dudhkoshi catchment (Fig. 2b), and 0.95 m yr$^{-1}$ during 2013–2015
(Mimeau et al., 2018) for Periche sub-catchment (Fig. 2b). The last two estimates compared well to those presented above.
Existing estimates (Chandel and Ghosh, 2021) of summer runoff (0.87 m yr$^{-1}$) and glacier runoff (0.76 m yr$^{-1}$) of Dudhkoshi
catchment were also consistent with our results. No such previous runoff estimates were available for Chandra catchment. The
estimated glacier contributions to runoff obtained here were largely consistent with the existing model studies from the region
(Nepal, 2016; Engelhardt et al., 2017; Mimeau et al., 2018; Azam et al., 2019; Chandel and Ghosh, 2021) when the differences
in fractional glacier cover were taken into account (supplementary Table S4).

The parameter-sensitivity analysis revealed that the absolute changes in summer runoff were less than $\sim 1.5\%$ for all the
parameters, except $Bf$ and $K_{slow}$ (Figs. 4b–4d). Slightly higher summer-runoff sensitivities (1.8–2.5 %) for the two longer
time scales $Bf$ and $K_{slow}$ became less than 1%, when the annual runoff was considered. The additional 80 simulations
where two parameters were perturbed simultaneously, obtained runoff changes almost equal to the sum of those obtained in
the corresponding pair of simulations with a single perturbed parameter (supplementary Fig. S5). A generally low parameter
sensitivity of the summer runoff implied that the present summer runoff estimates were relatively robust to the uncertainties
in these 11 glacio-hydrological model parameters (supplementary Table S2). The result of the ensemble of 200 models where
one of the 11 model parameters and the two calibration paramters were perturbed by $\pm25\%$ also produced reasonable mean
and uncertainty bands (Supplementary Fig. S6).

**4.3   Simulated glacier mass balance and its climate sensitivity**

The simulated glacier mass balance for Chandra and upper Dudhkoshi catchments over 1980–2018 were $-0.18\pm0.10$ and
$-0.37\pm0.04$ m w.e. yr$^{-1}$. These estimates were comparable to the existing geodetic observations within the uncertainties
(Fig. 6c; supplementary Table S3). The RMSE between modelled and observed mass balance of Chandra and upper Dudhkoshi
catchments were 0.10 and 0.11 m w.e. yr$^{-1}$, respectively.

The sensitivity of the modelled glacier mass balance to temperature was $-0.47\pm0.09$ and $-0.27\pm0.05$ m yr$^{-1}$ °C$^{-1}$
for Chandra and upper Dudhkoshi catchments, respectively. The corresponding precipitation sensitivities for these catchments
were $0.2\pm0.04$ and $0.05\pm0.02$ m yr$^{-1}$ for a 10% change in precipitation. These sensitivities were significant at $p < 0.01$
level. The previously reported mass-balance sensitivities at a regional scale (Shea and Immerzeel, 2016; Sakai and Fujita,
2017; Tawde et al., 2017; Wang et al., 2019) and for individual glaciers from the western and central Himalaya (Azam et al.,
2014; Wang et al., 2019; Sunako et al., 2019; Azam and Srivastava, 2020) spanned a wide range (supplementary Table S8). This
possibly reflected the corresponding differences of the climate setting, geometry, and topography of the glaciers studied, along
with underlying model assumptions, model calibration, input data sets, and so on. The mass-balance sensitivities obtained





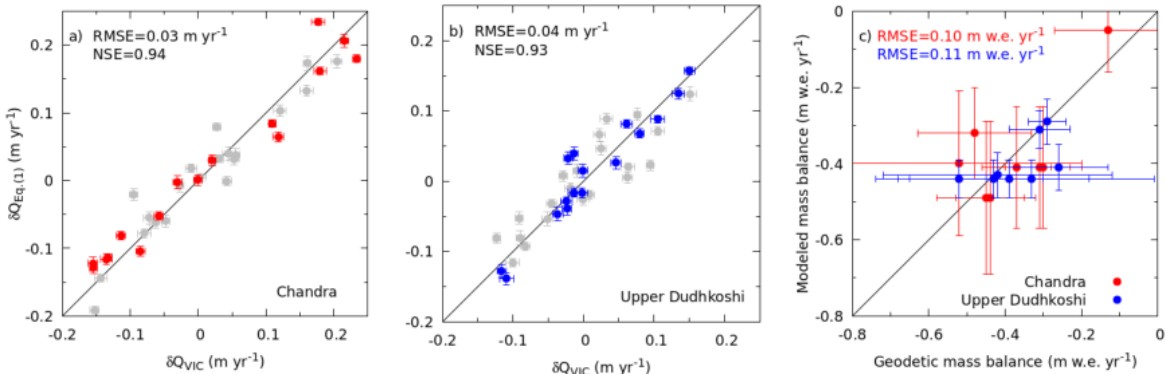

**Figure 6.** The summer runoff anomalies $\delta Q_{\mathrm{Eq.(1)}}$ as computed using the Eq. (3) are compared with those from the VIC model simulations $\delta Q_{\mathrm{VIC}}$, for (a) Chandra, and (b) upper Dudhkoshi catchments. The red (blue) solid circles are for Chandra (upper Dudhkoshi) catchment during the validation period 1980–1996. The gray solid circles denote data from the calibration period 1997–2018. (c) A comparison of the modelled glacier mass balance with the available regional-scale geodetic mass balance for Chandra (solid red circles) and upper Dudhkoshi (solid blue circles) catchments. The modeled values are over to the same period as that of the corresponding observed geodetic mass balance (supplementary Table S3). The solid gray line in each plot shows the 1:1 reference line.

in the present study were well within the above range. A relatively higher summer temperature sensitivity of the glaciers Chandra catchment compared to those of upper-Dudhkoshi was in apparent contradiction with an expected stronger influence
of temperature forcing on summer-accumulation type glaciers due to a conversion between snow and rain (Fujita, 2008; Kumar et al., 2019). However, apart from the precipitation seasonality, mass-balance sensitivity also depends on factors like glacier hypsometry such that a relatively weaker temperature-sensitivity of glaciers in summer-monsoon fed Dudhkoshi compared to that in winter-snow fed Chandra cannot be ruled out. In fact, a similar trend of mass-balance sensitivities over these two regions were also found in a regional-scale energy-balance model study (Sakai and Fujita, 2017).

**4.4 Climate sensitivities of catchment runoff**

A linear fit of the summer runoff anomalies to those of summer temperature and annual precipitation (Eq. (3)) during 1997–2018 worked well for both Chandra ($R^2$=0.92) and upper Dudhkoshi ($R^2$=0.93) catchments. These fits obtained respective temperature sensitivities of summer runoff $s_T$ of 0.12±0.01 and 0.12±0.03 m yr$^{-1}$ °C$^{-1}$ for Chandra and upper Dudhkoshi catchments, respectively. The corresponding best-fit $s_P$ were 0.39±0.03 and 0.47±0.06 m yr$^{-1}$ m$^{-1}$. These sensitivities were
all significant at $p < 0.01$ level. The estimated sensitivities for the two catchments were the same within the limits of uncertainty, and the corresponding percentage changes in runoff were also similar (supplementary Table S5). This may be a surprising feature given the contrasting precipitation regimes of the catchments. This issue is discussed later in the text.

The sensitivities computed over the calibration period (1997–2018) reproduced the variability of summer runoff over the validation period (1980–1996) reasonably well (Figs. 6a–6b) with RMSE < 0.04 m yr$^{-1}$ and NSE > 0.93. This also validated





the use of Eq. (3) to predict the interannual variability of summer runoff in these two catchments. The sensitivities reported here were also in line with the previous estimates from the region (Fujita and Sakai, 2014; Pokhrel et al., 2014; Azam and Srivastava, 2020) or elsewhere (Engelhardt et al., 2015; He, 2021) (supplementary Table S6).

During 1980–2018, the simulated summer runoff in Chandra and upper Dudhkoshi catchments varied in the range 0.86–1.33 and 0.55–0.98 m yr$^{-1}$, respectively. The respective ranges of summer temperature were 2.0–5.3 and 1.2–2.3°C, and those of

annual precipitation were 1.05–2.10 and 1.17–1.92 m yr$^{-1}$. As discussed before, the sensitivities estimated above are applicable within the above range of precipitation and temperature forcing. Note that in both the catchments, $s_P$ was significantly smaller than 1 m yr$^{-1}$ m$^{-1}$. This indicated an interannual change of the storage in the glaciers, and a change in evapotranspiration from the off-glacier area in response to the precipitation forcing (see Sects. 4.5 and 4.6).

### 4.5 Climate sensitivities of glacier runoff

The estimated temperature sensitivities of glacier runoff $s_T^{(g)}$ were 0.41±0.02 and 0.47±0.06 m yr$^{-1}$ °C$^{-1}$ for Chandra and upper Dudhkoshi catchments, respectively (significant at $p < 0.01$ level). The corresponding precipitation sensitivities $s_P^{(g)}$ were −0.12±0.08 and 0.00±0.02 m yr$^{-1}$ m$^{-1}$ (not significant at $p < 0.05$ level). A compilation of glacier runoff sensitivities (supplementary Table S7) indicated that the sensitivities reported by here were largely in line with those reported previously in the Himalaya (Fujita and Sakai, 2014; Chandel and Ghosh, 2021) and elsewhere (Anderson et al., 2010; Soruco et al.,

2015; Pramanik et al., 2018). Again, both the catchments had similar absolute values of $s_P^{(g)}$ and $s_T^{(g)}$ within the corresponding uncertainties. The corresponding percentage sensitivity values were also similar, except a somewhat higher percentage change in glacier runoff due to unit temperature change in upper Dudhkoshi catchment (supplementary Table S5).

Interestingly, summer runoff of both winter-accumulation type glaciers in Chandra catchment and summer-accumulation type glaciers in upper Dudhkoshi catchment was approximately independent of the corresponding precipitation variability.

This confirmed the general result, which was derived previously using a simple temperature-index model (Banerjee, 2022), that irrespective of the glacier chosen, the climate setting, or the model used, glacier runoff has a weak to no precipitation sensitivity.

In both the studied catchments, a positive precipitation anomaly did not translate into a higher summer runoff of the glaciers (Fig. 7). With increasing precipitation, the rainfall on glacier did not change, and snowmelt showed a very weak (Chandra) to

no (upper Dudhkoshi) increase (Fig. 7). This implied that a higher precipitation contributed mostly to a positive storage change (snow accumulation) on the glaciers. In addition, a higher snowcover and/or an association between higher-than-normal precipitation and lower mean temperature (not statistically significant) caused a decline in glacier melt, and amplified the changes in glacier storage change (Fig. 7). These effects combined to yield a nearly precipitation-insensitive glacier runoff in both the catchments. In contrast, a higher glacier melt with increasing mean summer temperature caused a relatively high temperature

sensitivity of $Q^{(g)}$ in both the catchments (Fig. 7). Here, the glaciers effectively acted as infinite reservoirs over an annual scale so that the meltwater volume was limited only by the available energy. A higher temperature implied a higher available energy, and thus a higher meltwater flux from the glaciers. These arguments were consistent with a high correlation ($r > 0.9, p < 0.05$) between the summer temperature and summer runoff of the glacierised parts for both the catchments (Fig. 7).





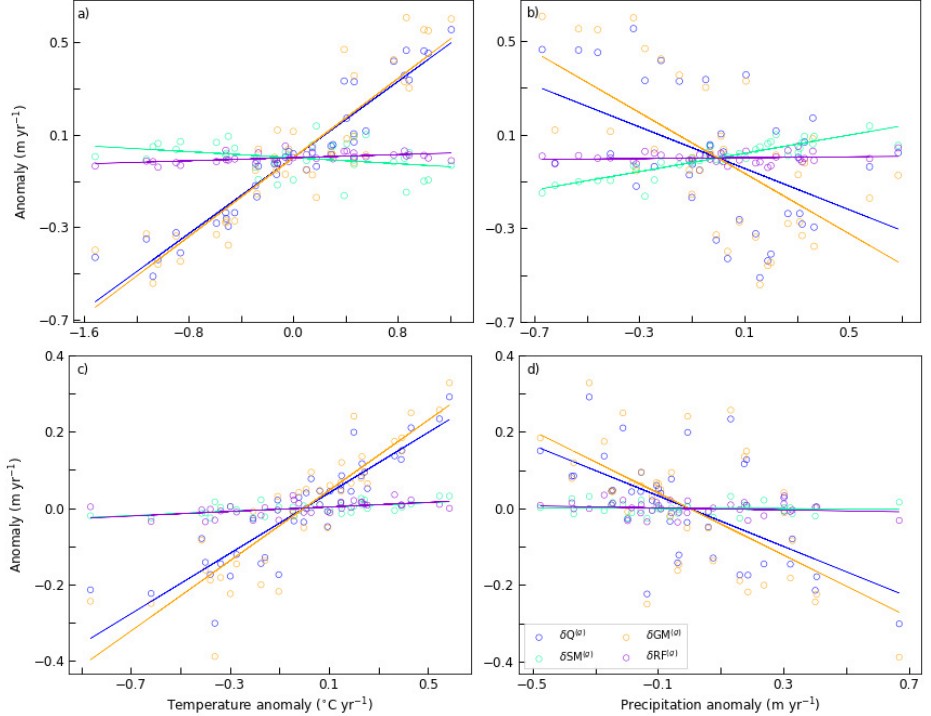

**Figure 7.** The anomalies of glacier runoff $\delta Q^{(g)}$, and its components, namely, snowmelt $\delta SM^{(g)}$, glacier ice melt $\delta GM^{(g)}$, and rainfall $\delta RF^{(g)}$ for the glacierised parts of the catchments are plotted as a unction of the corresponding temperature and precipitation anomalies: (a, b) for Chandra catchment, and (c, d) for upper Dudhkoshi catchment. The corresponding best-fit straight lines are also shown.

The negligible $s_P^{(g)}$ discussed above implied a stabilisation of the total runoff of the glacierised catchments against precipi-

tation variability, as the runoff contribution from the glacierised fraction $x$ was essentially independent of precipitation (Eq. 8). The magnitude of the precipitation sensitivity of catchment runoff $s_P$ is thus expected to decrease with the glacier fraction $x$. This stabilising effect (Banerjee, 2022) is consistent with a reported buffering of catchment runoff by glaciers during the extreme drought years across High Mountain Asia (Pritchard, 2019).

### 4.6 Climate sensitivity of runoff of the non-glacierised parts

In the non-glacierised parts of Chandra and upper Dudhkoshi catchments, $s_T^{(r)}$ of 0.02±0.01 and 0.03±0.04 m yr$^{-1}$ °C$^{-1}$ and $s_P^{(r)}$ of 0.56±0.04 and 0.59±0.07 m yr$^{-1}$ m$^{-1}$ were obtained, respectively. These sensitivities were all significant at $p < 0.01$ level. Again, both the catchments had similar absolute values of $s_P^{(r)}$ and $s_T^{(r)}$ within the corresponding uncertainties, and the corresponding percentage sensitivity values were similar (supplementary Table S5).

Compared to the sensitivities of glacier runoff, the climate sensitivities of the runoff from the non-glacierised parts showed

an exactly opposite trend. The summer runoff of the off-glacier areas were relatively insensitive to temperature anomalies, but





sensitive to precipitation anomalies (supplementary Fig. S8). Because of the presence of seasonal snow cover over the non-glacierised parts, a temperature dependence of the summer runoff may be expected. However, the total amount of snowmelt during the summer was limited by the supply of seasonal snow, and not by the available energy. This led to a weak response of the total summer runoff from the non-glacierised parts to temperature forcing. This argument was supported by the fact that the summer runoff from the non-glacierised parts were uncorrelated with summer temperature and strongly correlated with summer precipitation ($r > 0.9, p < 0.05$). Our results suggest that the precipitation changes in these two catchments caused comparable changes in surface runoff, groundwater/baseflow, and evapotranspiration (Supplementary Fig. S8). Consequently, about ~2/3rd of the precipitation anomaly translated to that of the total runoff. Interestingly, evapotranspiration anomalies in the glacier-free parts of Chandra (upper Dudhkoshi) were controlled by the summer temperature (precipitation) (Supplementary Fig. S8). This suggested a water-limited condition in the summer monsoon-fed upper Dudhkoshi catchment, and an energy-limited condition in the winter snow-fed Chandra catchment.

### 4.7 Implications of the estimated climate sensitivities

The above estimated climate sensitivities from glacierised and non-glacierised parts of the catchments suggested $s_P^{(g)} \approx 0$ and $s_T^{(r)} \approx 0$. Thus, Eqs. (3)–(10) can be simplified to the following approximate relations describing the response of the summer runoff to climate variability and change in these two catchments.

$$\delta Q \approx x s_T^{(g)} \delta T + (1-x) s_P^{(r)} \delta P, \tag{11}$$

$$\delta Q^{(g)} \approx x s_T^{(g)} \delta T, \tag{12}$$

$$\delta Q^{(r)} \approx (1-x) s_P^{(r)} \delta P, \tag{13}$$

$$\sigma_Q \approx \sqrt{x^2 s_T^{(g)2} \sigma_T^2 + (1-x)^2 s_P^{(r)2} \sigma_P^2}, \tag{14}$$

$$\Delta Q \approx x s_T^{(g)} \Delta T + (1-x) s_P^{(r)} \Delta P + \Delta x (Q_0^{(g)} - Q_0^{(r)}). \tag{15}$$

These simplified equation suggested that the key parameters that determined the climate response of these glacierised catchments to given climate forcing were $s_T^{(g)}$ and $s_P^{(r)}$. According to Eq. (11), the precipitation and temperature sensitivity of catchment runoff are essentially given by $(1-x)s_P^{(r)}$ and $x s_T^{(g)}$, respectively. As both the catchments had similar $s_P^{(r)}$, the corresponding $s_P$ were also similar with a slightly smaller value in Chandra catchment due to a higher fractional glacier cover there. On the other hand, a slightly higher $s_T^{(g)}$ in upper Dudhkoshi catchment, together with a slightly lower glacier cover there, led to similar $s_T$ in the two catchments. Below we discuss the implications of the above simplified linear-response formulae for the future changes in the mean summer runoff and its variability.

### 4.7.1 Summer runoff variability

Over the calibration period 1997–2018, Chandra and upper Dudhkoshi catchments had $\sigma_P$ of 0.22 and 0.15 m yr$^{-1}$, and $\sigma_T$ of 0.89 and 0.34 °C, respectively. These values, together Eq. (14), predicted $\sigma_Q$ of 0.13 and 0.08 m yr$^{-1}$ for the two catchments,



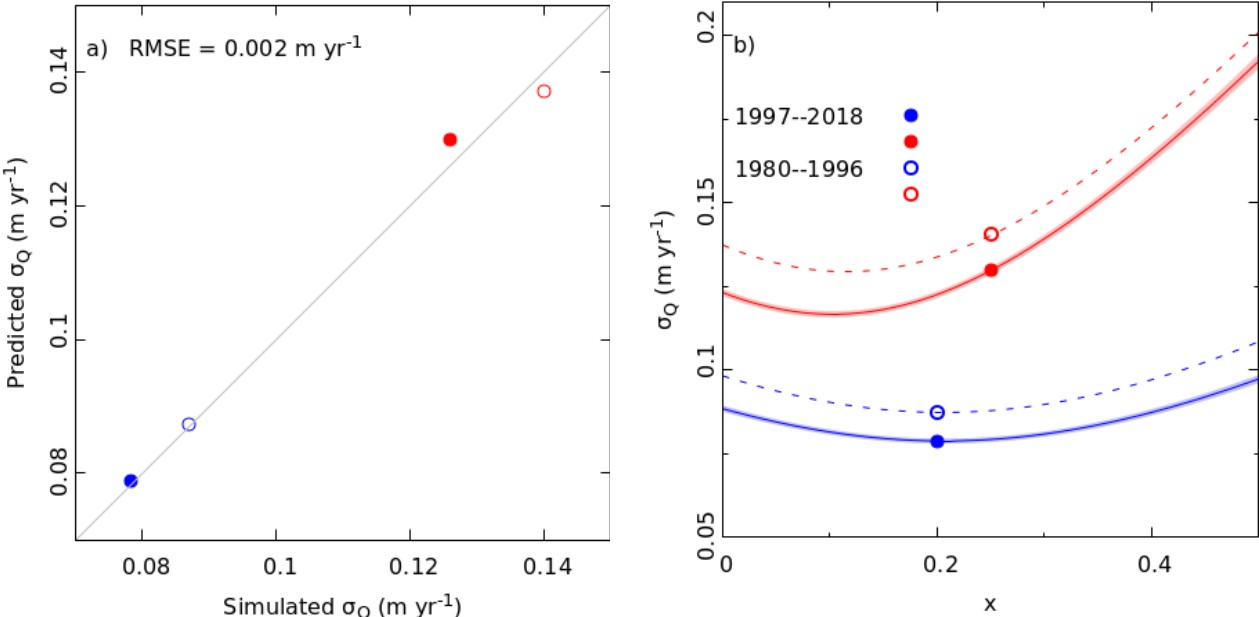

**Figure 8.** a) Predicted $\sigma_Q$ using Eq. (14) are compared with the corresponding simulated values for both the catchments. The solid and open circles denote data for the periods 1997–2018 and 1980–1996, respectively. Data for Chandra and upper Dudhkoshi catchments are shown with red and blue symbols, respectively. b) The solid (dashed) lines show $\sigma_Q(x)$ obtained using $\sigma_T$ and $\sigma_T$ values from 1997–2018 (1980–1996).

which equalled the corresponding values obtained directly from the simulated summer runoff (Fig. 8a). A corresponding close match was also obtained over the validation period of 1980-1996 (Fig. 8a).

Equation (14)) can also be used to predict the variation of $\sigma_Q$ in these catchments due to the shrinkage glacier cover if $\sigma_P$ and $\sigma_T$ were to remain unchanged. The shape of hyperbolic $\sigma_Q(x)$ curve for both the catchments (Fig. 8b) indicated that major

changes in runoff variability may not take place due to the expected glacier loss alone. However, possible changes in $\sigma_P$ and $\sigma_T$ may drive significant future changes of $\sigma_Q$ in these two catchments, as underlined by the difference between the simulated $\sigma_Q$ for the two catchments over the periods 1980–1996 and 1997–2018 (Fig. 8b).

### 4.7.2   Glacier-compensation curve

For a set of hypothetical catchments with different values of $x$, but similar $s_T^{(g)}$, $s_P^{(r)}$, $\sigma_T$ and $\sigma_P$, Eq. (14) implies that $\sigma_Q$ is a

hyperbolic function of $x$ (Fig. 8b). The runoff variability is high in the limit $x \to 0$ due to a precipitation sensitive off-glacier runoff with $\sigma_Q \approx (1-x)s_P^{(r)}\sigma_P$. In the opposite limit of $x \to 1$, $\sigma_Q$ is again high due to a high temperature sensitivity of glacier runoff, with $\sigma_Q \approx x s_T^{(g)}\sigma_T$. These two competing effects yield a minimum in $\sigma_Q$ at an intermediate value of $x$ (Fig. 8b)





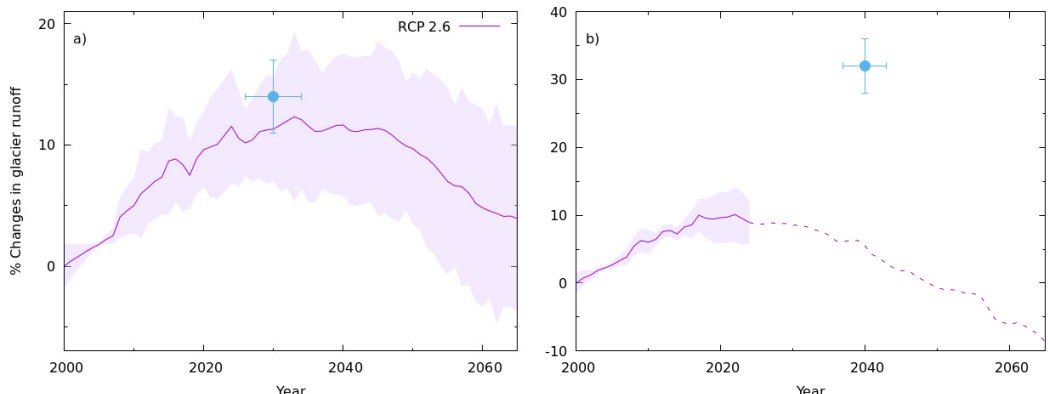

**Figure 9.** The 'peak water' due to future glacier changes predicted using Eq. (15) for (a) Chandra, and (b) upper Dudhkoshi catchments, respectively. The solid sky-blue dots represents the corresponding 'peak water' as reported in Huss and Hock (2018) for both the catchments. Dashed portion of the solid line in upper Dudhkoshi catchment indicate the corresponding temperature change beyond the calibration range of the catchment. See text for details.

(Banerjee, 2022). This nonmonotonic behaviour of runoff variability with $x$ is well known empirically (e.g., Chen and Ohmura, 1990), and is termed as glacier-compensation effect. The above theoretical explanation of the effect is consistent with a reported

strong correlation between runoff and precipitation (temperature) in the limit of small (extensive) glacier cover (Van Tiel et al., 2020). Note that while Eq. (14) suggests a hyperbolic glacier compensation curve, some of the existing studies used an empirical parabolic curve (e.g., Chen and Ohmura, 1990). As glacier cover shrink, the summer runoff from both the studied catchments is expected to become more sensitive to precipitation forcing (Eq. (14)).

    Chen and Ohmura (1990) suggested that the glacier-compensation curve can be utilised to estimate the change in $\sigma_Q$ as

glacier cover changes. However, recent model simulations indicated that a time-dependent glacier-compensation curve rules out such possibility (Van Tiel et al., 2020). This is consistent with Eq. (14), which indicates that apart from a changing glacier cover, the compensation curve (and thus $\sigma_Q$) can shift when $\sigma_P$ and/or $\sigma_T$ changes with time.

### 4.7.3    Changes in mean summer runoff and prediction of peak water

As discussed before, estimating the future changes mean summer runoff using Eq. (15) requires the changes in summer precip-

itation or temperature to be within the range of calibration (Sect. 3.1.3). Only for Chandra catchment, the optimistic RCP 2.6 scenario (supplementary Fig. S9), temperature change by $\sim$2050 was within the range of annual temperature over the period 1980–2018 and the present estimates of climate sensitivities could be used safely. The projected mean temperature changes of 1.1°C by 2050 under RCP 2.6 scenario were within the calibration range, and obtained a glacier runoff change of 0.27±0.03 m yr$^{-1}$ (assuming $x_0 = 1$ at 2000). This was comparable to the corresponding reported estimate of 0.25 m yr$^{-1}$ for the entire

Indus basin (Huss and Hock, 2018).





The predicted future changes of glacier runoff in Chandra catchments, using Eq. (15), reproduced the peak-water effect successfully (Fig. 9). The estimated peak water was $12\pm8\%$ of the present glacier runoff, and the estimated timing was $2033\pm7$. In Chandra catchment, the estimated peak-water in glacier runoff is expected to cause a $0.05$ m yr$^{-1}$ to rise in catchment runoff. This change may not be detectable, given the recent interannual variability of catchment runoff $\sigma_Q = 0.14$ m yr$^{-1}$. Note that

the above estimates are comparable to previously predicted a peak water of $14\pm3\%$ on $2030\pm4$ (Huss and Hock, 2018). It is encouraging that a simple climate-sensitivity based approach presented here could capture the peak-water effect in Chandra catchment as well as a state-of-the-art glacio-hydrological model (Huss and Hock, 2018). Note that for Chandra catchment, our simulated recent glacier runoff, the initial glacier cover, and geodetic mass balance used for calibration were similar to the corresponding values used by Huss and Hock (2018) for the Indus basin.

In upper Dudhkoshi catchment, we estimated a peak water of $10\pm4\%$ of the present glacier runoff, and the estimated timing was $2022\pm4$ (Fig). This estimated peak water was significantly smaller and quicker compared to that of Huss and Hock (2018). This inconsistency may be related to possible extrapolation errors as, the projected temperature changes crossed the range of interannual variability by 2024. Moreover, there were several difference between the two models in this region, which may contribute to the above mismatch. The RGI 4 glacier inventory used by Huss and Hock (2018) had 25% higher glacier cover

in Ganga basin compared to RGI 6 used here. Also the authors calibrated their model using a geodetic mass-balance record which was twice as negative as the median of the eight geodetic mass balance records used here. Also, the present estimates of glacier runoff in upper Dudhkoshi catchment was almost half of that reported by Huss and Hock (2018) for Ganga basin. The above differences likely led to a corresponding large difference in the modelled climate sensitivities of glacier runoff between the present study and that of Huss and Hock (2018) in this region.

## 5   Summary and conclusions

In this paper, we simulate the runoff of Chandra (western Himalaya) and upper Dudhkoshi (eastern Himalaya) catchments over 1980–2018, using the VIC model augmented with a temperature-index glacier-melt module. Calibrating two model parameters using a Bayesian method that simultaneously fits the available summer runoff and decadal-scale geodetic glacier mass balance, our simulation obtained a good match with both observations and the existing model results. The simulated climate sensitivities

of summer runoff to temperature and precipitation forcing in the catchments reveal some interesting patterns. The precipitation sensitivities of the summer runoff of the non-glacierised parts of the catchments are high, but those of the glacierised parts are negligible. In contrast, the temperature sensitivities of summer runoff of glaciers are high, but those of the non-glacierised parts are negligible. As a consequence, the temperature sensitivity of the glacier runoff and the precipitation sensitivity of the off-glacier runoff are critical determinants of the future changes of summer runoff and its variability in these two catchments.

Despite the limitations like calibration with a limited dataset, the use of a simple temperature-index ice-melt module, the static-glacier assumption, ignoring the effects of supraglacial debris cover and avalanching, and so on, the present study brings out some interesting similarities in the climate sensitivities of two glacierised Himalayan catchments with contrasting climate regimes.



*Data availability.* All the observed hydro-meteorological data, and basic simulation results will be made public.

*Author contributions.* AB conceived the study, and developed the theoretical framework. SL ran the simulations. SL and AB did the data analysis. AB wrote the manuscript with inputs from SL. AS, PS, and MT did the field measurements at Chandra catchment, and analysed the field data. All the authors participated in the discussions, and edited the manuscript.

*Competing interests.* The authors have no competing interests to declare.

*Acknowledgements.* The study is funded under the HiCOM initiative by ESSO-NCPOR, MoES (grant number NCAOR/2018/HiCOM/04).
SL acknowledges INSPIRE Research fellowship (grant no IF 170526). AB acknowledges support from MoES grant no MoES/PAMC/H&C/79/2016-PC-II. We thank the French Agence Nationale de la Recherche (references ANR-09-CEP-0005-04/PAPRIKA and ANR-13-SENV-0005-03/PRESHINE) for sharing the hydro-meteorological data in the upper Dushkoshi catchment. We greatly benefited from discussions with Subimal Ghosh, Pradeep P Mujumdar, and Raghu Murtugudde. Vikram Chandel, Ela Chawla, Arijeet Dutta, and Sneha Potghan helped with VIC model setup and input data.



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
