# Peer review of "Climate sensitivity of the summer runoff of two glacierised Himalayan catchments with contrasting climate"

_Hydrology and Earth System Sciences, 2022_

## Author Comment (AC1)

Authors' **response** to the **comments** by Anonymous Referee #1

In this paper, the authors quantified the sensitivity of summer runoff to precipitation and temperature changes in two glacierized Himalayan catchments with contrasting climate based on a hydrological model. This study is well prepared, and the results are meaningful. However, there are still some questions needing to be clarified. Here are the Details:
We thank the reviewer for the careful comments.

Lines 48-49: This expression should be careful. In my opinion, the simulation is not as same as the observations. The observed data can represent the reality at a point scale but is hard to obtain, especially in the high mountain regions, while the simulation can systematically analyze for a basin-wide scale over a long period.
While we agree that observation and simulation both have their limitations and strengths, the point we make here is that when observations are scant (eg, in the studied catchments), model runs may be the only way to understand the sensitivities. Also, note that the discharge is a point measurement, it naturally integrates over the whole catchment.

Figure 1. The boundaries of the two study basins should be highlighted on the map.
The boundaries were not visible clearly enough due to the scale of the location map, which is why detailed boundaries were provided separately in Fig 2.

Lines 65-66: This sentence is inaccurate. As I see, the annual temperature of Chandra (-55â„ƒ) is lower than Upper Dudhkoshi (-4.7â„ƒ). In addition, the glacierized fraction of Chandra (0.25) is higher than Upper Dudhkoshi (0.20), and the former glacier area is more than two times the latter. These differences are significant and have a large impact on the glaciohydrology. So please revise it.
We shall modify the text in the revised text.

Lines 165-166: How to calculate the glacier area change? There is only the glacier mass balance change data in the supplementary Figure S3.
The present glacio-hydrological model did not consider the changes in glacier area, which was relatively small over the study period. We discussed this point in L 165-169.

In addition, have the model considered the compensation of snow and transforms into ice?
The snow module (Andreadis et al., 2009) in VIC does not consider compaction/firnification. Thus we have only snow or ice on the glacier surface. However, the effect of the snow metamorphosis is included in the paramterisation of the albedo change.

Supplementary Figure S1: How to deal with the observed data gaps?
In this figure (and also in Fig S2), we only considered months where the gaps were less than one week. Each point in the resultant plots is the mean over at least 2 (5) years in Chandra (upper Dudhkoshi). We shall add these details in the revised caption.

Line 220-225: I think the temperature before the ablation season can also influence the glacier melt and snowmelt since it controls the distribution of rainfall and snowfall in the accumulation season. Especially in the Chandra basin, where most precipitation occurs in the winter. Thus, I suggest that the authors should add a temperature sensitivity experiment before the ablation season.
Given that the mean temperature of Chandra catchment during Dec-Apr is less than –5°C, the effect of warming on snow-rain partitioning is unlikely to be important. However, we shall explore this possibility.

Lines 312-317: How to define the glacier runoff in this paper and what is the difference between it and the glacier ice loss?
Glacier runoff is the sum of the snow melt, ice melt, and rainfall on the glacerised part of the catchment as defined in L 159–160. The glacier mass balance (= snowfall-snowmelt-icemelt) is defined in L 160–161.

Moreover, the results show that the glacier runoff contribution to the total summer runoff in upper Duhkoshi is higher than that in Chandra basin while the glacier cover in upper Duhkoshi is lower than that in Chandra and the former summer precipitation is much higher than the latter, which seems contradictory.

Thank you for pointing out this inconsistency. There was due to a calculation error. The revised glacier ice melt contributions to the summer runoff are 36 and 31% in Chandra and upper Dudhkoshi catchments, respectively. We shall correct the error in the revised text. This trend is also visible in Fig S7 where Chandra shows a larger annual glacier runoff.

Lines 351-354: How does the glacier hypsometry affect the mass-balance sensitivity, and how is this factor considered in the model?

The hypsometry of the glacier cover in the catchments were explicitly included in the VIC+DDF model used to compute glacier mass balance (L 154–155).

Lines 377-379: In my opinion, the results vary from different studies, I think the authors should discuss the reason for the difference among the studies at different basins rather than describe it as "largely in line with".

We shall revise the sentence and add some relevant discussion.

Figure 7: The color scheme is too blurry to distinguish.

We shall revise the figure.

Line 389: The precipitation increased while the rainfall on glacier did not change, why? Please clarify it.

The higher precipitation contributed mostly to a positive storage change, i.e., snow accumulation on the glaciers (L 390).

Lines 464-466: The RCP2.6 scenario data has been used in this paper, but has no introduction (e.g. which general circulation model has been selected and the evaluation of the projected data).

Here we have used the data from Huss and Hock 2018 and Kraaijenbrink et al 2017. We referred the reader to these papers for the relevant details (L 279-281).

**References**

Andreadis, K. M., Storck, P., and Lettenmaier, D. P. (2009). Modeling snow accumulation and ablation processes in forested environments. Water resources research, 45(5).

Huss, M., and Hock, R. (2018). Global-scale hydrological response to future glacier mass loss. Nature Climate Change, 8(2), 135-140.

Kraaijenbrink, P. D., Bierkens, M. F. P., Lutz, A. F., and Immerzeel, W. W. (2017). Impact of a global temperature rise of 1.5 degrees Celsius on Asia's glaciers. Nature, 549(7671), 257-260.

---

## Author Comment (AC2)

Authors' **response** to the **comments** by Anonymous Referee #2

We thank the reviewer for the careful comments and criticisms. Reviewer #2 has recommended rejection, essentially based on two criticisms: 1) our model does not represent reality well enough, and 2) our conclusions are not significant. Before providing a detailed response to the reviewers' comments, we present our response on the above two points.

1) Our model is a reasonable one in terms of process representations, and also in terms of data used for calibration. This is evident when the present model is compared with all the 18 cited glacio-hydrological model studies from the region (Table R1 below). The calibration using both runoff and glacier mass balance data simultaneously (sect 3.2.3), and the detailed comparisons of modelled sensitivities of mass-balance (sect 4.3) and runoff (sect 4.2) with existing studies, do not reveal any inconsistencies in our results. Moreover, our process-based explanations for the pattern of sensitivities reported (sect 4.5) support that the effects may be real.

**Table R1:** A comparison of the characteristics of the glacio-hydrological studies over the High Mountain Asia, with the present study:   The entries denote the number of model studies having a particular feature.

| | Heavily glacierised Himalayan catchments (6 models in total) | Sparsely glacierised Himalayan catchments (8 models in total) | Regional studies covering High Mountain Asia (4 models in total) | This study |
|---|---|---|---|---|
| Glacier baseflow | Yes: 0 No: 6 | Yes: 0 No: 6 | Yes: 0 No: 6 | No |
| Glacier storage | Ignored:4 Linear reservoir: 2 | Ignored: 3 Linear reservoir: 5 | - | Linear reservoirs |
| Sub-debris ice melt | 4 | 2 | 1 | No |
| Snow redistribution model | 1 | 1 | 1 | No |
| Icemelt model | DDF: 5 Energy balance: 1 | DDF: 7 Energy balance: 1 | DDF: 2 Energy balance: 1 | DDF |
| Runoff observations | >5yrs: 2 <5yrs: 4 | >5yrs: 6 <5yrs: 2 | - | >5yrs: Dudhkoshi <5yrs: Chandra |
| Mass-balance observations | Yes: 5 No: 1 | Yes: 2 No: 6 | Yes: 4 No: 0 | Yes |
| Input precipitation correction | Yes: 5 No: 1 | Yes: 5 No: 1 | Yes: 4 No: 0 | Yes |
| Number of calibration parameters | 3 to 5: 1 =<2: 4 | 3 to 5: 2 <=2: 0 | - | 2 |

2) We are not aware of any systematic and focused studies of climate sensitivities of the glacierisied Himalayan catchments. In particular, the solely precipitation-independent (temperature-independent) glacier (off-glacier) runoff in these catchments has neither been pointed out nor been explained in terms of the underlying processes before. The corresponding implications for the change and variability of runoff have not been studied as well. Thus, the results presented may be considered significant.

This is a well written manuscript on a very relevant topic regarding the contribution of streamflow generated in the glacier-covered part of a catchment to catchment-scale water resources. It uses two contrasting glacierised

Himalayan catchments, one of which is winter-precipitation dominated, Chandra (the western Himalaya), and the other one summer-precipitation dominated, upper Dudhkoshi (the eastern Himalaya). For these catchments, climate sensitivities of simulated streamflow is obtained by regressing the simulated variability of streamflow to the one its meteorological drivers. The used model is a the Variable Infiltration Capacity (VIC) model, augmented with a glacier melt module.

We thank the reviewer for the critical comments.

The analysis is model-based; , the used precipitation-glacier-melt-streamflow model is very simple for the glacier-covered catchment part; as far as I see, it sums up the ice melt and the snowmelt (and rainfall) and routes it through a single (or perhaps two, unclear) linear reservoir, i.e. the corresponding streamflow response has a single time scale stemming from icemelt and snowmelt and no baseflow, thus the model can most likely not simulate a water carry-over effect from month to month for the glacier part. This model structure might have a different impact on the estimated sensitivities for the different analysed catchments.

We beg to differ with the reviewer's opinion here.

The baseflow for the Himalayan glaciers was ignored assuming the negligible permeability of the bedrock. As table R1 shows the same assumption was made in most, if not all, of the available studies (Table R1). We do include baseflow for the off-glacier area (including lateral basins/moraines around the glaciers) (Fig 3).

About glacier storage, all the existing studies either ignore the process or use a reservoir model as used in this study (Table R1). Note that the slow (monthly) timescale in glacier runoff is due to the snow storage which is incorporated in our two-reservoir model (L 160).

Furthermore, we do not have information on how large the (ignored) debris cover is nor on how important snow redistribution is, we simply know that it is ignored.

Only 4-7% of the studied catchments consist of debris-covered ice (Scherler et al., 2018), and their influence on the streamflow was ignored. In fact, a simple inclusion of the melt inhibiting effects of the debris layer, as done in some of the existing model studies (Table R1), may not necessarily lead to improved description of reality. For example, the strong melt enhancements at the ice-cilffs/ponds on the debris-covered surface (e.g., Miles et al., 2022) are usually ignored in the above models. The available estimates of the extent and thickness estimates have large uncertainties as well.

The variability of wind and gravity driven snow-redistribution in the rugged Himalayan topography are also difficult to capture in any coarse-scale model like the present one.

We shall add these details/arguments in the revised version.

Only two parameters of the hydrological model are calibrated, the ones that affect the water balance the most strongly (melt factor for ice and precipitation scaling factor). The calibration is on streamflow and glacier mass balance; there is an empirical weight factor to combine the performance with respect to both quantities; despite a clear lack of giving any formal statistical framework, the parameter estimation approach is called a Bayesian Inference.

High Himalayan catchments, like the one studied here, are data-sparse. So we use a minimal set of two calibration parameters to avoid over-fitting (L 172). All the other parameters were assigned reasonable values, and the corresponding sensitivity was shown to be small (L 210–213). We believe calibrating a small subset of the model parameters is a common compromise in hydrological modelling (eg, Table R1).

The empirical factor was included to give similar weightage to both the objectives while optimising (L 197-198). The factor of half can be absorbed in $\sigma_b$, then it is equivalent to using the 80 percentile of the set of uncertainty values (Table S3) instead of the median. We shall expand the relevant discussion, and include this point.

The present approach of using uniform prior distributions for the two fit-parameters (L 185), using Bi-variate Gaussian distribution for the residues associated with the two (independent) observed datasets (L 189), and obtaining a posterior probability distribution for the models (shown explicitly in Fig 4a, 4b), is a Bayesian approach. Here we have followed the formulation used in Rounce et al. (2020), Werder et al. (2020), etc. from the glaciology literature which we are familiar with.

Accordingly, I am rather skeptical about the added value of this model study; I think that this is essentially a modelling exercise without clear indications that it actually corresponds to how nature reacts;
We beg to differ due to the following reasons:
- Given the importance of the data-sparse catchments for us, modelling exercises like the present one are important.
- We have utilised all the available hydro/metero/glacio-hydrological data (even though limited) to improve the representation of natural processes.
- We incorporate reasonable representations of the most important glacio-hydrological processes, following the available studies in the literature (Table R1).
- We have acknowledged the model limitations, which are often shared with most of the studies from the region as shown (Table R1). (though we admit that the reiewer has pointed out a few points where more discussion would have been beneficial).
- The detailed discussions of the consistency of our result with existing modeling studies (runoff: L 318–324, regional glacier mass balance: Table S3, catchment runoff sensitivities: Table S6, glacier runoff sensitivities: Table S7, glacier mass balance sensitivities: Table S8) support the validity of our result.
- The detailed discussions of the underlying processes driving the sensitivity pattern reported, provide support in favour of the validity of the results (Sect 4.5–4.6).

moreover, the conclusion is very general with new insights that can be inferred from general process knowledge such as e.g. the sentence "the temperature sensitivity of the glacier runoff and the precipitation sensitivity of the off-glacier runoff are critical determinants of the future changes of summer runoff and its variability in these two catchments". I therefore recommend rejection of this version.
Again we beg to differ as,
- We are not aware of any study in the Himalaya where the above pattern of sensitivity has been discussed based on either process understanding or quantitative modeling of glacierised catchments.
- We are not aware of any studies where the implication of the above property on the variability and change in runoff of any Himalayan catchment have been analysed.

The work could become more valuable if it was more critical about the value of the model, if it discussed what we miss with the simplifications and if it provided more insights in what we can learn from the two different types of catchments.
The focus of the work is to understand the pattern of runoff sensitivities of the two Himalayan catchments and the role of glaciers in that. Given the existing glacio-hydrological models for Himalayan catchment, the present model appears to be a rather reasonable model (Table R1). However, as the reviewer has aptly pointed out, some of the assumptions could have been explained better, and there are a few missing details. We shall make appropriate corrections.

Detailed comment:
what do you mean by runoff? there are usages of this term where it does not include groundwater-fed baseflow; accordingly: if we mean total flow leaving a catchment, we might want to use streamflow;
We are happy to review the usage of runoff, and replace it with streamflow wherever needed. However, we have encountered similar usage of runoff in cited literature (e.g. Khadka et al 2014, He and Pang 2015, Bhattacharya et al 2019).

it needs to be very clear also what is meant by "glacier runoff": runoff generated in the glacier-covered part of the catchment? including baseflow? Including runoff from lateral moraines that are not glacier covered?
Glacier runoff is unambiguously defined in L 159-160. We shall add here that baseflow is ignored (see TableR1, L 164). Lateral moraines, lateral basins, etc are not usually considered as part of the glacier.

Methods, calibration: I do not think that 5% or 10% error on summer streamflow observations is a realistic value; since summer is the period of high flow, this values is certainly, much higher;
The uncertainties in the observed runoff could not be accessed. We assumed twice the value that was reported in another Himalayan catchment (L 193). We shall acknowledge the possibility that the error could be higher.

furthermore, in the chosen formulation, the error should correspond to the total model error and not just to the observational error (see e.g. an Bayesian inference paper by Dmitri Kavetski);
We agree that ideally $\sigma^2_{tota} = \sigma^2_{model} + \sigma^2_{observation}$. However, as we don't have estimates of model errors, incorporating those using, say, an uniform prior (eg, Rounce et al (2020)), would expand the dimension of the model space to 4. To avoid that complication, we ignored the model error (e.g., Warder et al 2020). We shall discuss this point in the revised text.
The paper referred to, discusses methods for considering noise in individual rainfall events. In this paper, we do acknowledge that input rainfall is a major source of error, and employ a simple constant multiplicative factor to correct for it. Given that we do intend to capture the total summer runoff and not the catchment response to the individual rain events, this simplification may not be entirely unreasonable, and is in line with the existing studies (Table R1).

the Bayesian formulation for the mass balance is based on very few obs. values;
We have used all the available mass balance estimates (L 194).

what assumption do you make about the distribution of the residuals for streamflow and mass balance? i.e. what motivates the chosen form of the likelihood?
They are assumed to be from a Bivariate Gaussian distribution (eq 2), which is a commonly used assumption (e.g., Raounce et al 2020, Warder et al 2020). We also assume them to be uncorrelated, which we have confirmed using the sampled posterior distribution. We shall add these details.

How did you compute the posterior (you did not use any sampling method that would yield a sample for the posterior; I guess you did some kind of rescaling?)
The posterior distribution is the truncated bivariate Gaussian shown in Fig 4a,b (We apologise for mislabeling the colormap label as $P(\theta|d)$, it should be $P(d|\theta)$ ). Since we have a two-dimensional model space, we could map out the distribution over the entire domain with 11X29 model runs, and no sampling algorithm was not needed.

Methods, other: i) what is the used temporal time step of the VIC model on the glacier part ?
As stated in L 116, we ran the VIC model at hourly time steps in both glacerised and nonglacierised parts of the catchment.

why is it reasonable to keep the bias correction constant in space? I would expect that biases depend on elevation?
We corrected ERA5 precipitation with an elevation-independent scale factor, which was calibrated using the observed runoff and the decadal regional glacier mass balance. Due to complex topography in the Himalaya, even within a single grid box, an elevation-dependent correction may not be a better representation of the precipitation pattern (e.g., Johnson and Rupper 2020). Please see the discussion in L 105-108.

Why does the glacier melt model not use the energy-balance approach?
Due to scarcity of field data, we chose a minimal DDF model for the ice melt, following most of the hydrological studies in the region (see Table R1).

Is the glacier melt coded by the authors of the study or someone else?
We do not understand the relevance of the question.
We have given due credit to any piece of code used in this paper that is not written by us, if that is what is being referred to.

Methods: the computation of glacier mass balance sensitivity is not clear to me; did you run the model with modified precipitation input?
As stated in L 248–249, we did not perturb the precipitation, but used the interannual variability of mass balance to compute the sensitivities (just as we do for the runoff). We shall make this explicit in the revised text.

Methods: how did you compute the deltaP and deltaT values (anomalies)?
As stated in L 236–238 : "*We also considered the runoff from glacierised part of the catchments $Q^{(g)} = Q_0^{(g)} + \delta Q^{(g)}$, and that from the non-glacierised part of the catchments $Q^{(r)} = Q_0^{(r)} + \delta Q^{(r)}$ . Here, the notations $Q_0$ and $\delta Q$ denote the long-term mean and the anomaly for a given year, respectively.*"

Results: please reword "the present calibration strategy resolved the equifinality problem that is usually encountered while calibrating glacio-hydrological models using only discharge data"; using two data sets does not remove equifinality; you built a single performance metric with an empirical factor to sum up two performance measures and then you report only the best value; it does not mean that there is no Equifinality
We apologise for the confusing sentene.
We used equifinality in a restricted sense where the same discharge is produced in a heavily glacierised catchment, say, by either melting more ice (via a higher DDF) or by having a higher precipitation (via a higher $\alpha_P$). This is just as discussed in the cited ref. Azam and Srivastava (2020), where the modeled runoff fitted the observations equally well even as the relative contributions of precipitation and glacier melt varied. This is the specific problem that is resolved with our approach.
In the revised text we shall calrify that we are talking about the equifinality, i.e., the family of non-unique solutions to the problem of fitting the observed runoff, in the context of the 2-d model space (DDF, $\alpha_P$) that we (or Azam and Srivastava (2020) etc.) have considered.

Fig. 1: the legend (not the caption) should also include what the dashed lines are
We shall update the legend.

Fig. 3: the glacier scheme is probably wrong, the text states that there are two linear reservoirs, rainfall is missing
We shall revise the flowchart to correct this error. Rainfall was indeed considered as given in the text (L 159).

Fig. 6: what is the y-axis (equation 1 is the calibration equation, something is wrong?)?
We shall revise the figure.

Fig. 7: should be improve, I cannot see much about the circles
We shall revise the figure.

Table 1: is there snowfall occurring in summer and if yes, what is the amount of sommer snowfall? Caption could say how summer is defined (it is in the text though);
The monthly snowfall is shown in Fig 1c.
Snowfall constitutes 14% and 5% of summer precipitation in Chandra and upper Dudhkoshi catchments, respectively.

how high is ET?
ET is about ~⅓ of the annual precipitation in these catchments (Fig S7).

The following reference which is certainly relevant is missing: van Tiel et al :
https://hess.copernicus.org/articles/25/3245/2021/; probably their review on glacier

We shall add the above reference in the revised version.

**References**

Azam, M. F., and Srivastava, S. (2020). Mass balance and runoff modelling of partially debris-covered Dokriani Glacier in monsoon-dominated Himalaya using ERA5 data since 1979. Journal of Hydrology, 590, 125432.

Bhattacharya, T., Khare, D., and Arora, M. (2019). A case study for the assessment of the suitability of gridded reanalysis weather data for hydrological simulation in Beas river basin of North Western Himalaya. Applied Water Science, 9(4), 1-15.

He, R., and Pang, B. (2015). Sensitivity and uncertainty analysis of the Variable Infiltration Capacity model in the upstream of Heihe River basin. Proceedings of the International Association of Hydrological Sciences, 368, 312-316.

Johnson, E., and Rupper, S. (2020). An examination of physical processes that trigger the albedo-feedback on glacier surfaces and implications for regional glacier mass balance across high Mountain Asia. Frontiers in Earth Science, 8, 129.

Khadka, D., Babel, M. S., Shrestha, S., and Tripathi, N. K. (2014). Climate change impact on glacier and snow melt and runoff in Tamakoshi basin in the Hindu Kush Himalayan (HKH) region. Journal of Hydrology, 511, 49-60.

Miles, E. S., Steiner, J. F., Buri, P., Immerzeel, W. W., and Pellicciotti, F. (2022). Controls on the relative melt rates of debris-covered glacier surfaces. Environmental Research Letters, 17(6), 064004.

Rounce, D. R., Khurana, T., Short, M. B., Hock, R., Shean, D. E., and Brinkerhoff, D. J. (2020). Quantifying parameter uncertainty in a large-scale glacier evolution model using Bayesian inference: application to High Mountain Asia. Journal of Glaciology, 66(256), 175-187.

Scherler, D., Wulf, H., and Gorelick, N. (2018). Global assessment of supraglacial debris-cover extents. Geophysical Research Letters, 45(21), 11-798.

Werder, M. A., Huss, M., Paul, F., Dehecq, A., and Farinotti, D. (2020). A Bayesian ice thickness estimation model for large-scale applications. Journal of Glaciology, 66(255), 137-152.

---

## Author Response (AR1)

We thank the editor and reviewers for their insightful comments. Before providing our point-by-point response, first we list out three important changes based on the comments.

(1) We incorporated the model error while calibrating the model (Eq. 2 and Sect. 3.2.3) and calculating the uncertainties of simulated runoff (Sect 3.2.4), climate sensitivities, and so on (Sect. 3.2.4).
(2) We tested the statistical significance of the model fits to runoff and mass balance observations. (Sect. 4.1 L330–333, and Fig S7).
(3) We have expanded the discussion of the limitations of the glacier model like not including the debris-cover effects (Sect. 3.2.2 L169–178).

Authors' **response** to the **comments** by Editor

I read your paper, the reviewer comments and your responses with great care and interest. While both reviewers agree on the general importance of the scope of your study, they were also rather critical.

Particularly reviewer 2 is "rather skeptical about the value" of the study, and raised serious doubts about the realism of the proposed model and the statistical soundness of the calibration approach. While I acknowledge the quality of your responses, I also share several important concerns, which make very major revisions necessary. Beside the considerable number of points, that need a better explanation (partly in line with your responses), the following points need special attention:
Thank you for your suggestions. Please see our response below.

As you cite the work of (Beven and Freer, 2001) you should be precise: Equifanility means that several parameter sets perform in an acceptable manner!
We have rewritten the discussion as, (L186–190) "*It may also suffer from equifinality issue (Beven and Freer, 2001; Jost et al., 2012), where more than one parameter combinations reproduce the observed runoff. For example, in glacierised catchments, the same runoff output can be generated by models which use different combinations of the precipitation scale factor and DDF (Azam and Srivastava, 2020). These models will, however, yield different relative contributions of glaciers to the total runoff, and obtain different climate sensitivities.*"

Quantification of the related predictive model uncertainty (and thus the model error) implies essentially to define a model acceptance threshold a-priory, and which allows construction of an uncertainty band (which yields the model error) based on the acceptable parameter sets.
We have now updated the conditional probability by including both model and observational errors (Eq 2 and Sect. 3.2.3). We also tested the statistical significance of the model fits to the observations relative to those in a randomly chosen model (Sect. 4.1 and Fig S7). We computed the uncertainty bands for all relevant quantities using the conditional probability, L226: "*All the relevant quantities were computed for all the 319 models in the weighted ensemble, and the corresponding  weighted standard deviations were used to obtain the 2σ uncertainties.*"

Temperatures sensitivities of glacier melt need to be judged compared to the model error (and observation error), to evaluate their significance.
Please see the previous response.

When doing so, you should acknowledge that the chosen likely-hood, the NSE, requires a careful application in areas with such a strong seasonality of discharge. The null model is not the simple overall average discharge, but the averaged annual cycle thereof (which is a deterministic signal). Hence, it is absolutely necessary to compare the model against the performance of the averaged annual cycle of stream discharges, to judge how much it explains compared to this simple null model.

As clearly stated in L207 ("*Here, $Q_i$ was the weekly summer runoff, and the summation was over all the years with observed runoff data.*"), we used weekly runoff values to compute the RMSE of runoff (Eq 2), thereby incorporating its seasonality.

In addition, we now tested the statistical significance of the fits to observed runoff and glacier mass balance as obtained in the most probable model, as described in L330–333: "*To test the statistical significance of the above fits, we computed the probability of having RMSEs of runoff and mass balance equal to or smaller than those in the best-fit model, when the entire model space is sampled uniformly (supplementary Fig. S7). For both discharge and glacier mass balance, these probabilities were 0.03 or smaller in both the catchments, indicating that the fits were significant at $p < 0.05$ level.*"

I also think that RC1 made, among others, a valid point of adding a temperature sensitivity experiment before the ablation season.
As the mean temperature of catchments during Dec-Apr is less than $-5°C$, the effect of $1°C$ warming on snow-rain partitioning or on snowmelt is not expected to be important. We confirm that the non-summer temperature perturbations do not affect the summer runoff in both catchments.

Authors' **response** to the **comments** by Anonymous Referee #1

In this paper, the authors quantified the sensitivity of summer runoff to precipitation and temperature changes in two glacierized Himalayan catchments with contrasting climate based on a hydrological model. This study is well prepared, and the results are meaningful. However, there are still some questions needing to be clarified. Here are the Details:
We thank the reviewer for the encouraging comments. Our response to the detailed comments are provided below.

Lines 48-49: This expression should be careful. In my opinion, the simulation is not as same as the observations. The observed data can represent the reality at a point scale but is hard to obtain, especially in the high mountain regions, while the simulation can systematically analyze for a basin-wide scale over a long period.
While we agree that observation and simulation both have their limitations and strengths, the point we make here is that when observations are scant (eg, in the studied catchments), model runs may be the only way to understand the sensitivities. Also, we note that while discharge is measured at a point, it naturally integrates over the whole catchment.

Figure 1. The boundaries of the two study basins should be highlighted on the map.
The boundaries were not visible clearly enough due to the scale of the location map, which is why detailed boundaries were provided separately in Fig 2.

Lines 65-66: This sentence is inaccurate. As I see, the annual temperature of Chandra (-55â„f) is lower than Upper Dudhkoshi (-4.7â„f). In addition, the glacierized fraction of Chandra (0.25) is higher than Upper Dudhkoshi (0.20), and the former glacier area is more than two times the latter. These differences are significant and have a large impact on the glaciohydrology. So please revise it.
Thank you for the comment. We now modified the discussions as (L68–72) "*The catchment area of upper Dudhkoshi is approximately half of Chandra catchment. The glacierised fraction in Chandra catchment is 20% higher than that of upper Dudhkoshi. The mean annual temperature is $0.8°$ C lower in Chandra catchment compared to upper Dudhkoshi catchment. However, the former has a more pronounced seasonality with a warmer summer and a cooler winter (Fig. 1d). Chandra catchment has a somewhat higher annual and summer runoff. Some important characteristics of the two catchments are compared in Table 1.*"

Lines 165-166: How to calculate the glacier area change? There is only the glacier mass balance change data in the supplementary Figure S3.
The present glacio-hydrological model did not consider the changes in glacier area, which was relatively small over the study period (Table 1). We already discussed this point in L 179–181: "*We assumed a static glacier cover*

*here as the observed percentage loss of glacier area over the simulation period was small (1–5 % decade⁻¹) for both the catchments (Table 1). Biases due to such a static-glacier approximation were found to be small for another glacierised Himalayan catchment over the same period (Azam and Srivastava, 2020).*"

In addition, have the model considered the compensation of snow and transforms into ice?
Our static glacier model has only snow or ice on the glacier surface. The snow module (Andreadis et al., 2009) in VIC does not consider compaction/firnification explicitly, but the evolution of albedo due to snow metamorphosis is included in it.

Supplementary Figure S1: How to deal with the observed data gaps?
In this figure (and also in Fig S2), we only considered months where the gaps were less than one week. Each point in the resultant plots is the mean over at least 2 (5) years in Chandra (upper Dudhkoshi).
We have now clarified this in the figure caption: "*Here, we only considered months where the observed data gaps were less than one week. Each point in the plots represents the mean over at least 2 (5) years in Chandra (upper Dudhkoshi) catchment.*"

Line 220-225: I think the temperature before the ablation season can also influence the glacier melt and snowmelt since it controls the distribution of rainfall and snowfall in the accumulation season. Especially in the Chandra basin, where most precipitation occurs in the winter. Thus, I suggest that the authors should add a temperature sensitivity experiment before the ablation season.
Given that the mean temperature of Chandra catchment during Dec-Apr is less than –5°C, the effect of warming on snow-rain partitioning is unlikely to be important. Indeed, when we looked at the sensitivity of summer runoff to non-summer temperature fluctuations, we found them to be insignificant.

Lines 312-317: How to define the glacier runoff in this paper and what is the difference between it and the glacier ice loss?
Glacier runoff is the sum of the snow melt, ice melt, and rainfall on the glacierised part of the catchment as already defined in L 163–164: "*The snow melt, ice melt, and rainfall on the glaciers routed through two linear reservoirs (Hannah and Gurnell, 2001) with reservoir constants $K_{fast}$ and $K_{slow}$ , obtained the glacier runoff (e.g., Radić and Hock, 2014).*"
The mass balance is also defined in L164–165: "*Catchment-wise glacier mass balance was computed by subtracting the total ice and snow melt from the total snowfall over the glacierised parts.*"

Moreover, the results show that the glacier runoff contribution to the total summer runoff in upper Duhkoshi is higher than that in Chandra basin while the glacier cover in upper Duhkoshi is lower than that in Chandra and the former summer precipitation is much higher than the latter, which seems contradictory.
Thank you for pointing out this inconsistency. This was due to a calculation error. The revised glacier ice melt contributions to the summer runoff are 36 and 31% in Chandra and upper Dudhkoshi catchments, respectively. We have corrected the error in the revised text (L341–342). This trend was visible in Fig S6 where Chandra showed a larger annual glacier runoff.

Lines 351-354: How does the glacier hypsometry affect the mass-balance sensitivity, and how is this factor considered in the model?
The static hypsometry (obtained from the DEM using the RGI glacier boundaries) of the glacier cover in the catchments was explicitly included in the VIC+DDF model used to compute the glacier mass balance (L 158–159: "*On the glacierised grids of each catchment, a separate VIC model computed the snow melt and snow-covered fraction of each elevation band (Fig. 3). A temperature-index model (Hock, 2003) obtained the ice melt over the corresponding snow-free areas.*" ).

Lines 377-379: In my opinion, the results vary from different studies, I think the authors should discuss the reason for the difference among the studies at different basins rather than describe it as "largely in line with".

We modified the text L389–399: "*The reported temperature sensitivities of summer runoff over the Himalaya were in the range between 5 to 27 % of summer runoff change per ◦ C warming, and the precipitation sensitivities of summer runoff were between -0.6 to 16 % of summer runoff due to 10% changes in P (Fujita and Sakai, 2014; Pokhrel et al., 2014; Azam and Srivastava, 2020) (supplementary Table S6). The previously reported temperature and precipitation sensitivities of summer runoff outside the Himalaya were in the range between 9 to 24 % of summer runoff per ◦ C warming, and 2 to 7 % of summer runoff due to 10% changes in P (Engelhardt et al., 2015; He, 2021), respectively. The temperature and precipitation sensitivities of summer runoff obtained in the present study, 11–14 % of summer runoff per ◦ C warming and 6–9 % of summer runoff due to 10% changes in P, respectively, were within the above ranges. It appears that the differences in climate sensitivities of runoff obtained in different studies are mostly due to the corresponding differences in glacier fraction of the catchments studied, as there is a monotonic variation of the sensitivities with glacier fractions (supplementary Table S6).*"

Figure 7: The color scheme is too blurry to distinguish.
We have revised the figure.

Line 389: The precipitation increased while the rainfall on glacier did not change, why? Please clarify it.
The higher precipitation contributed mostly to a positive storage change, i.e., snow accumulation on the glaciers. This was already discussed in L422: "*This implied that a higher precipitation contributed mostly to a positive storage change (snow accumulation) on the glaciers.*".

Lines 464-466: The RCP2.6 scenario data has been used in this paper, but has no introduction (e.g. which general circulation model has been selected and the evaluation of the projected data).
Here we have used the relevant data from Huss and Hock 2018, and Kraaijenbrink et al 2017. We have referred the readers to the relevant papers L302: "*The corresponding temperature projections were obtained from available estimates for the western and eastern Himalaya, respectively (supplementary Fig.S8 of Kraaijenbrink et al. (2017)).*"

**References**

Andreadis, K. M., Storck, P., and Lettenmaier, D. P. (2009). Modeling snow accumulation and ablation processes in forested environments. Water resources research, 45(5).

Huss, M., and Hock, R. (2018). Global-scale hydrological response to future glacier mass loss. Nature Climate Change, 8(2), 135-140.

Kraaijenbrink, P. D., Bierkens, M. F. P., Lutz, A. F., and Immerzeel, W. W. (2017). Impact of a global temperature rise of 1.5 degrees Celsius on Asia's glaciers. Nature, 549(7671), 257-260.

Authors' **response** to the **comments** by Anonymous Referee #2

We thank the reviewer for the careful comments and criticisms. Reviewer #2 has recommended rejection, essentially based on two criticisms: 1) our model does not represent reality well enough, and 2) our conclusions are not significant. Before providing a detailed response to the reviewers' comments, we present our response on the above two points.
1) Our model is a reasonable one in terms of process representations, and also in terms of data used for calibration. This is evident when the present model is compared with all the 18 cited glacio-hydrological model studies from the region (Table R1 below). The calibration using both runoff and glacier mass balance data simultaneously (Sect 3.2.3), and the detailed comparison of modelled sensitivities of mass-balance (Sect 4.3) and runoff (Sect 4.2) with existing studies, do not reveal any inconsistencies in our results. Moreover, our process-based explanations for the pattern of sensitivities reported (Sect 4.5) support that the effects reported may be robust.

**Table R1:** A comparison of the characteristics of the glacio-hydrological studies over the High Mountain Asia, with the present study: The entries denote the number of model studies having a particular feature.

| | Heavily glacierised Himalayan catchments (6 models in total) | Sparsely glacierised Himalayan catchments (8 models in total) | Regional studies covering High Mountain Asia (4 models in total) | This study |
|---|---|---|---|---|
| Glacier baseflow | Yes: 0 No: 6 | Yes: 0 No: 6 | Yes: 0 No: 6 | No |
| Glacier storage | Ignored:4 Linear reservoir: 2 | Ignored: 3 Linear reservoir: 5 | - | Linear reservoirs |
| Sub-debris ice melt | 4 | 2 | 1 | No |
| Snow redistribution model | 1 | 1 | 1 | No |
| Icemelt model | DDF: 5 Energy balance: 1 | DDF: 7 Energy balance: 1 | DDF: 2 Energy balance: 1 | DDF |
| Runoff observations | >5yrs: 2 <5yrs: 4 | >5yrs: 6 <5yrs: 2 | - | >5yrs: Dudhkoshi <5yrs: Chandra |
| Mass-balance observations | Yes: 5 No: 1 | Yes: 2 No: 6 | Yes: 4 No: 0 | Yes |
| Input precipitation correction | Yes: 5 No: 1 | Yes: 5 No: 1 | Yes: 4 No: 0 | Yes |
| Number of calibration parameters | 3 to 5: 1 =<2: 4 | 3 to 5: 2 <=2: 0 | - | 2 |

2) We are not aware of any systematic and focused study of climate sensitivities of the glacierisied Himalayan catchments. In particular, the solely precipitation-independent (temperature-independent) glacier (off-glacier) runoff in these catchments has neither been pointed out nor been explained in terms of the underlying processes before. The corresponding implications for the change and variability of runoff have not been studied as well. Thus, the results presented may be considered significant.

This is a well written manuscript on a very relevant topic regarding the contribution of streamflow generated in the glacier-covered part of a catchment to catchment-scale water resources. It uses two contrasting glacierisedHimalayan catchments, one of which is winter-precipitation dominated, Chandra (the western Himalaya), and the other one summer-precipitation dominated, upper Dudhkoshi (the eastern Himalaya). For these catchments, climate sensitivities of simulated streamflow is obtained by regressing the simulated variability of streamflow to the one its meteorological drivers. The used model is a the Variable Infiltration Capacity (VIC) model, augmented with a glacier melt module.

The analysis is model-based; , the used precipitation-glacier-melt-streamflow model is very simple for the glacier-covered catchment part; as far as I see, it sums up the ice melt and the snowmelt (and rainfall) and routes it through a single (or perhaps two, unclear) linear reservoir, i.e. the corresponding streamflow response has a single time scale stemming from icemelt and snowmelt and no baseflow, thus the model can most likely not simulate a water carry-over effect from month to month for the glacier part. This model structure might have a different impact on the estimated sensitivities for the different analysed catchments.

We beg to differ with the reviewer's opinion here that our glacier model is inadequate. As shown in Table R1, the present model compares favourably with those used in the existing studies in the region.

The baseflow for the Himalayan glaciers was ignored here assuming the negligible permeability of the bedrock. As table R1 shows the same assumption was made in most, if not all, of the available studies (Table R1). We do include baseflow for the off-glacier area (includes lateral basins/moraines around the glaciers) (Fig 3).

About glacier storage, all the existing studies either ignore it or used the same or similar reservoir model as used in this study (Table R1). Note that the slow (monthly) timescale in glacier runoff is due to the snow storage which is already incorporated in our two-reservoir model (L164).

Furthermore, we do not have information on how large the (ignored) debris cover is nor on how important snow redistribution is, we simply know that it is ignored.
Thank you for pointing this out. We have now expanded the relevant discussion, L169–178: " *…as only 4–7% of the studied catchments consist of debris-covered ice (Scherler et al, 2018). A simple inclusion of the melt inhibiting effects of the debris layer (e.g., Azam and Srivastava, 2020) may not necessarily lead to improved estimation of subdebris melt. For example, the strong melt enhancements at the ice-cilffs/ponds on the debris-covered surface (e.g. Miles et al, 2022) are often ignored in these models. The available estimates of the extent (e.g. Herreid and Pellicciotti, 2020) and thickness estimates (e.g. Rounce et al., 2018) have large uncertainties as well. Here, we verified that a simplified sub-debris melt scheme (Azam and Srivastava, 2020), which does not consider the variation of debris thickness, induced only small (~3%) insignificant changes in the summer runoff compared to the corresponding uncertainties (~10%). The effect of wind and gravity driven snow-redistribution in the rugged Himalayan topography are also difficult to capture in any coarse-scale model like the present one. As we are calibrating the observed mass balance of glaciers and catchment runoff, it may take care of the effects of these two factors to some extent.*"

Only two parameters of the hydrological model are calibrated, the ones that affect the water balance the most strongly (melt factor for ice and precipitation scaling factor). The calibration is on streamflow and glacier mass balance; there is an empirical weight factor to combine the performance with respect to both quantities; despite a clear lack of giving any formal statistical framework, the parameter estimation approach is called a Bayesian Inference.
High Himalayan catchments, like the one studied here, are data-sparse. So we use a minimal set of two calibration parameters to avoid over-fitting (L 186). All the other parameters were assigned reasonable values, and the corresponding sensitivity was shown to be small (L 352–358). We believe calibrating a small subset of the model parameters is a common compromise in hydrological modelling (eg, Table R1).

We modified the conditional probability (Eq 2) by including both the model and observation errors (Eq 2) and by removing the empirical weight of 1/2. Please refer to the revised Sect. 3.2.3 for the details. The present approach of using uniform prior distributions for the two fit-parameters (L 191), using a Bi-variate Gaussian distribution for the residues associated with the two (independent) observed datasets (L 206), and obtaining a posterior probability distribution for the models (shown explicitly in Fig 4a, 4b), is a Bayesian approach to the best of our understanding.

Accordingly, I am rather skeptical about the added value of this model study; I think that this is essentially a modelling exercise without clear indications that it actually corresponds to how nature reacts;
We beg to differ due to the following reasons:
 ● Given the importance of the data-sparse catchments for us, modelling exercises like the present
one are important.
 ● We have utilised all the available hydro/metero/glacio-hydrological data (even though limited) to
improve the representation of natural processes.
 ● We incorporate reasonable representations ofmost of the important glacio-hydrological processes,
following the available studies (Table R1).

● We have acknowledged the model limitations, which are shared with most of the previous model studies from the region as shown (Table R1). We expanded the discussion on the model limitations in Sect. 3.2.2.

 ● The detailed discussions of the consistency of our result with existing modeling studies (runoff: L 344–351, regional glacier mass balance: Table S3, catchment runoff sensitivities: Table S6, glacier runoff sensitivities: Table S7, glacier mass balance sensitivities: Table S8) support the validity of our result.

 ● The detailed discussions (Sect 4.5–4.6) of the underlying processes driving the sensitivity pattern reported, provide support in favour of the validity of the results.

moreover, the conclusion is very general with new insights that can be inferred from general process knowledge such as e.g. the sentence "the temperature sensitivity of the glacier runoff and the precipitation sensitivity of the off-glacier runoff are critical determinants of the future changes of summer runoff and its variability in these two catchments". I therefore recommend rejection of this version.

Again, we beg to differ as,

 ● We are not aware of any study in the Himalaya where the above pattern of sensitivity has been quantified using glaciohydrological modeling and explained in terms of the underlying processes.

 ● We are not aware of any studies where the implication of the above property on the variability and change in runoff of any Himalayan catchment have been analysed.

The work could become more valuable if it was more critical about the value of the model, if it discussed what we miss with the simplifications and if it provided more insights in what we can learn from the two different types of catchments.

Given the existing glacio-hydrological models for Himalayan catchment, the present model appears to be a rather reasonable model (Table R1). In addition, calibration, validation, uncertainty estimates, sensitivity to model parameters, and model limitations are documented and discussed in detail. In the revised version, we updated our calculations by incorporating model errors and expanded the discussion of model assumptions (Sect 3.2.2) further based on your suggestions.

The focus of the work is to understand the pattern of runoff sensitivities of the two Himalayan catchments and the role of glaciers in that. We also discussed the similarities of hydrological response properties between the two catchments with contrasting climate ( L384–386: "*The estimated sensitivities for the two catchments were the same within the limits of uncertainty, and the corresponding percentage changes in runoff were also similar (supplementary Table S5). This may be a surprising feature given the contrasting precipitation regimes of the catchments.*", and

L415–416: "*Interestingly, summer runoff of both winter-accumulation type glaciers in Chandra catchment and summer-accumulation type glaciers in upper Dudhkoshi catchment was approximately independent of the corresponding precipitation variabilities.*").

Detailed comment:

what do you mean by runoff? there are usages of this term where it does not include groundwater-fed baseflow; accordingly: if we mean total flow leaving a catchment, we might want to use streamflow;

Thank you for point this out. To avoid any ambiguity, we clarified at the outset that (L47): "*Also, runoff of a catchment implies the streamflow at the catchment outlet, and glacier (off-glacier) runoff denotes the contribution of the glacierised (glacier-free) areas of the catchment to the streamflow.*"

it needs to be very clear also what is meant by "glacier runoff": runoff generated in the glacier-covered part of the catchment? including baseflow? Including runoff from lateral moraines that are not glacier covered?

Glacier runoff is unambiguously defined in L163–164.

We have added that baseflow is ignored (see TableR1, and L167: "*We did not consider any baseflow contribution from the glacierised parts assuming the negligible permeability of the bedrock*").

Lateral moraines, lateral basins, etc are not usually considered as part of the glacier and are not included in the RGI 6 glaciers boundaries used.

Methods, calibration: I do not think that 5% or 10% error on summer streamflow observations is a realistic value; since summer is the period of high flow, this values is certainly, much higher;

The error in the observed runoff could not be accessed, therefore we now take a conservative estimate of 20% (L213–215: "*Here, $\sigma_Q$ was taken to be ~20% of the mean summer runoff of the catchments, which is a conservative estimate given the previously reported 5% error discharge measured using the same method for other Himalayan catchments (e.g., Singh et al., 2005).*").

furthermore, in the chosen formulation, the error should correspond to the total model error and not just to the observational error (see e.g. an Bayesian inference paper by Dmitri Kavetski);

We have now incorporated the model error, please refer to Sect 3.2.3 (Eq 2, L212) for details.

the Bayesian formulation for the mass balance is based on very few obs. values;

We have used all the available mass balance estimates for both the catchments, as already mentioned in L209.

what assumption do you make about the distribution of the residuals for streamflow and mass balance? i.e. what motivates the chosen form of the likelihood?

They are assumed to be from a Bivariate Gaussian distribution (Eq 2), which is a commonly used assumption (e.g., Rounce et al 2020, Warder et al 2020). We also assume them to be uncorrelated, which we have confirmed for the entire model space.

How did you compute the posterior (you did not use any sampling method that would yield a sample for the posterior; I guess you did some kind of rescaling?)

The posterior distribution is the truncated bivariate Gaussian shown in Fig 4a,b. Since we have a two-dimensional model space, we could map out the distribution over the entire domain simply with 11X29 model runs, and no sampling algorithm was not needed.

Methods, other: i) what is the used temporal time step of the VIC model on the glacier part ?

As stated in L120–121, we ran the VIC model at hourly time steps in both glacierised and nonglacierised parts of the catchment.

why is it reasonable to keep the bias correction constant in space? I would expect that biases depend on elevation?

We corrected ERA5 precipitation with an elevation-independent scale factor, which was calibrated using the observed runoff and the decadal regional glacier mass balance. Due to the complex topography in the Himalaya, even within a single grid box, an elevation-dependent correction may not be a better representation of the precipitation pattern (e.g., Johnson and Rupper 2020). We already clarified this in L 109–112: "*In some of the existing studies in the region, an elevation-dependent precipitation scaling has also been employed (e.g., Azam et al., 2019). However, as an elevation-dependent correction may potentially introduce additional uncertainties (e.g., Johnson and Rupper, 2020), we preferred a constant αp keeping the number of calibration parameters to a minimum.*"

Why does the glacier melt model not use the energy-balance approach?

Due to the scarcity of field data, we chose a minimal DDF model for the ice melt, following most of the hydrological studies in the region (see Table R1).

Is the glacier melt coded by the authors of the study or someone else?

We do not understand the relevance of the question.
We have given due credit to any piece of code used in this paper that is not written by us.

Methods: the computation of glacier mass balance sensitivity is not clear to me; did you run the model with modified precipitation input?

As already stated in L 270, we did not perturb the precipitation but used the interannual variability of mass balance to compute the sensitivities (just as we do for the runoff), L 270: "*Apart from the sensitivities of summer runoff, we also computed the precipitation and temperature sensitivities of glacier mass balance using the corresponding simulated interannual variability over the period of 1980–2018.*

Methods: how did you compute the deltaP and deltaT values (anomalies)?
As already stated in L 258–260: "*We also considered the runoff from glacierised part of the catchments $Q^{(g)} = Q_0^{(g)}+\delta Q^{(g)}$, and that from the non-glacierised part of the catchments $Q^{(r)} = Q_0^{(r)} + \delta Q^{(r)}$. Here, the notations $Q_0$ and $\delta Q$ denote the long-term mean and the anomaly for a given year, respectively.*"

Results: please reword "the present calibration strategy resolved the equifinality problem that is usually encountered while calibrating glacio-hydrological models using only discharge data"; using two data sets does not remove equifinality; you built a single performance metric with an empirical factor to sum up two performance measures and then you report only the best value; it does not mean that there is no Equifinality
We apologise for the confusing sentence. We used equifinality in a restricted sense. We now rephrased the discussion as L186–190: "*It may also suffer from equifinality issue (Beven and Freer, 2001; Jost et al., 2012), where more than one parameter combinations reproduce the observed runoff. For example, in glacierised catchments, the same runoff output can be generated by models which use different combinations of the precipitation scale factor and DDF (Azam and Srivastava, 2020). These models will, however, produce a different relative contribution to the runoff from the glaciers and the off-glacier parts, and obtain different climate sensitivities.*"

Fig. 1: the legend (not the caption) should also include what the dashed lines are
The dashed line is already explained in Fig 1 caption: "*...Area-elevation distribution of the catchments (solid lines + solid symbols), and that of the glacierised parts (dashed lines + solid symbols). c) Mean monthly precipitation (solid lines + solid symbols), along with the monthly snowfall (dashed lines + solid symbols)...*"

Fig. 3: the glacier scheme is probably wrong, the text states that there are two linear reservoirs, rainfall is missing
We have revised the flowchart and corrected this error. Rainfall was considered as stated in the text (L163).

Fig. 6: what is the y-axis (equation 1 is the calibration equation, something is wrong?)?
We have revised the figure.

Fig. 7: should be improve, I cannot see much about the circles
We have revised the figure.

Table 1: is there snowfall occurring in summer and if yes, what is the amount of sommer snowfall? Caption could say how summer is defined (it is in the text though);
The monthly snowfall is shown in Fig 1c.
Snowfall constitutes 14% and 5% of summer precipitation in Chandra and upper Dudhkoshi catchments, respectively.

how high is ET?
ET is about ~⅓ of the annual precipitation in these catchments (Fig S7).

The following reference which is certainly relevant is missing: van Tiel et al : https://hess.copernicus.org/articles/25/3245/2021/; probably their review on glaciermodelling is also relevant: https://wires.onlinelibrary.wiley.com/doi/10.1002/wat2.1483
We have now added the above references in L432 and L193.

**References**

Johnson, E., and Rupper, S. (2020). An examination of physical processes that trigger the albedo-feedback on glacier surfaces and implications for regional glacier mass balance across high Mountain Asia. Frontiers in Earth Science, 8, 129.

Rounce, D. R., Khurana, T., Short, M. B., Hock, R., Shean, D. E., and Brinkerhoff, D. J. (2020). Quantifying parameter uncertainty in a large-scale glacier evolution model using Bayesian inference: application to High Mountain Asia. Journal of Glaciology, 66(256), 175-187.

Werder, M. A., Huss, M., Paul, F., Dehecq, A., and Farinotti, D. (2020). A Bayesian ice thickness estimation model for large-scale applications. Journal of Glaciology, 66(255), 137-152.

---

## Referee Report (RR1)

This is a well written manuscript on a very relevant topic regarding the contribution of streamflow generated in the glacier-covered part of a catchment to catchment-scale water resources. It uses two contrasting glacierisedHimalayan catchments, one of which is winter-precipitation dominated, Chandra (the western Himalaya), and the other one summer-precipitation dominated, upper Dudhkoshi (the eastern Himalaya). For these catchments, climate sensitivities of simulated streamflow is obtained by regressing the simulated variability of streamflow to the one its meteorological drivers. The used model is a the Variable Infiltration Capacity (VIC) model, augmented with a glacier melt module.

The analysis is model-based; , the used precipitation-glacier-melt-streamflow model is very simple for the glacier-covered catchment part; as far as I see, it sums up the ice melt and the snowmelt (and rainfall) and routes it through a single (or perhaps two, unclear) linear reservoir, i.e. the corresponding streamflow response has a single time scale stemming from icemelt and snowmelt and no baseflow, thus the model can most likely not simulate a water carry-over effect from month to month for the glacier part. This model structure might have a different impact on the estimated sensitivities for the different analysed catchments.

We beg to differ with the reviewer's opinion here that our glacier model is inadequate. As shown in Table R1, the present model compares favourably with those used in the existing studies in the region.

*I would like to apologize for imprecise reading; the use of two linear reservoirs was indeed stated in the original paper, but I overlooked it. This could be made clearer in the model scheme. Besides: are glacier melt, snowmelt and rainfall summed up before entering the two linear models or are they routed separately? The reference probably says it but would be good to have here)*

(..)

Only two parameters of the hydrological model are calibrated, the ones that affect the water balance the most strongly (melt factor for ice and precipitation scaling factor). The calibration is on streamflow and glacier mass balance; there is an empirical weight factor to combine the performance with respect to both quantities; despite a clear lack of giving any formal statistical framework, the parameter estimation approach is called a Bayesian Inference.

High Himalayan catchments, like the one studied here, are data-sparse. So we use a minimal set of two calibration parameters to avoid over-fitting (L 186). All the other parameters were assigned reasonable values, and the corresponding sensitivity was shown to be small (L 352–358). We believe calibrating a small subset of the model parameters is a common compromise in hydrological modelling (eg, Table R1).

We modified the conditional probability (Eq 2) by including both the model and observation errors (Eq 2) and by removing the empirical weight of 1/2. Please refer to the revised Sect. 3.2.3 for the details. The present approach of using uniform prior distributions for the two fit-parameters (L 191), using a Bi-variate Gaussian distribution for the residues associated with the two (independent) observed datasets (L 206), and obtaining a posterior probability distribution for the models (shown explicitly in Fig 4a, 4b), is a Bayesian approach to the best of our understanding.

Thanks for having clarified and corrected the conditional probability formulation, accounting now for the model residuals (instead of simply the observational error) and removing the factor ½. Because of the omission of the model residuals and the empirical factor, the chosen approach was unclear and appeared to me an ad-hoc calibration criterion; it is clearer now. As far as I see, it is not mentioned in the revised version that you make an explicit assumption about the distribution of the model residuals (only in the response), which might be good for non-expert readers. It is indeed common to make the normality assumption for discharge residuals; I am not sure if there are any examples of assumption about mass balance residuals (this is a detail of course).

Furthermore, I now understand that you sampled the full 2D parameter space but kept the corresponding residual variance constant to a fixed value to compute the full posterior probability. In my experience, the residual variance is often sampled along with the parameters. The chosen solution is pragmatic (especially given the few mass balance observations) but not the standard procedure.

(..)

moreover, the conclusion is very general with new insights that can be inferred from general process knowledge such as e.g. the sentence "the temperature sensitivity of the glacier runoff and the precipitation sensitivity of the off-glacier runoff are critical determinants of the future changes of summer runoff and its variability in these two catchments". I therefore recommend rejection of this version.
Again, we beg to differ as,
  ● We are not aware of any study in the Himalaya where the above pattern of sensitivity has been quantified using glaciohydrological modeling and explained in terms of the underlying processes.
  ● We are not aware of any studies where the implication of the above property on the variability and change in runoff of any Himalayan catchment have been analysed.

I still maintain my comment. The take home messages of the analysis read as follows (full relevant text of the paper copied below):
  "The precipitation sensitivities of the summer runoff of the non-glacierised parts of the catchments are high, but those of the glacierised parts are negligible. In contrast, the temperature sensitivities of summer runoff of glaciers are high, but those of the non-glacierised parts are negligible. As a consequence, the temperature sensitivity of the glacier runoff and the precipitation sensitivity of the off-glacier runoff are critical determinants of the future changes of summer runoff and its variability in these two catchments. "

  We could have guessed that glacier runoff is temperature sensitive (in a model that uses only temperature as driver for melt) and that the off-glacier parts are sensitive to precipitation input. The abstract gives some more interesting text but the conclusion is definitively not yet there.

(..)

Why does the glacier melt model not use the energy-balance approach?
  Due to the scarcity of field data, we chose a minimal DDF model for the ice melt, following most of the hydrological studies in the region (see Table R1).

Ok, but the snowmelt model of VIC already uses an energy balance approach, i.e. the energy balance approach is set-up for snow melt, this means that all data is available? Or do I get this wrong? Accordingly, my question still holds: why is the energy balance approach feasible for the snow on the glacier but not for the ice melt under the snow? If this is simply for practical reasons (coding), it is perhaps worth mentioning.

Is the glacier melt coded by the authors of the study or someone else?
We do not understand the relevance of the question.
We have given due credit to any piece of code used in this paper that is not written by us.

I asked the question because it is unclear if the code is available, which is always interesting for other users.

Methods: the computation of glacier mass balance sensitivity is not clear to me; did you run the model with modified precipitation input?
As already stated in L 270, we did not perturb the precipitation but used the interannual variability of mass balance to compute the sensitivities (just as we do for the runoff), L 270: "*Apart from the sensitivities of summer runoff, we also computed the precipitation and temperature sensitivities of glacier mass balance using the*

*corresponding simulated interannual variability over the period of 1980–2018.*

I still do not understand. Please not that my question arose due to the sentence following the one above.

270     Apart from the sensitivities of summer runoff, we also computed the precipitation and temperature sensitivities of glacier mass balance using the corresponding simulated interannual variability over the period of 1980–2018. The precipitation sensitivity of glacier mass balance was defined to be the mass-balance change due to a 10% change in precipitation following the convention used in the literature (e.g., Wang et al., 2019).

how high is ET?
ET is about ~⅓ of the annual precipitation in these catchments (Fig S7).

(..)

---

## Author Response (AR2)

We again thank both the reviewers and the editor for their careful comments and suggestions. Below we provide our point-by-point response in **blue** text color.

Authors' **response** to the **comments** by Reviewer 1

The authors have thoughtfully revised the paper and I think it can be published.
Thank you for the positive comment on the manuscript.

But I think the figures and tables should be improved before publication. Here are the details:
1. The legends of all the shapes should be pointed out in the figure and the legend for all subplots should better place outside rather than inside an individual subplot (e.g., Fig 1b-d, Fig 7, Fig, 8, Fig 9,), making the reader easier to get the information from the figure.
We changed the legend's position as suggested.

2. The font in Table 1 should be uniform.
We used the format prescribed in HESS latex template (https://www.hydrology-and-earth-system-sciences.net/submission.html#templates) to prepare the table.

Authors' **response** to the **comments** by Reviewer 2

Re-review: I carefully read the response and the revised paper. I have some additional comments. Below, the red parts are my original comments. The black are the author's answers. In yellow are my new comments.
We thank the reviewer for the careful comments. Please see our replies below.

This is a well written manuscript on a very relevant topic regarding the contribution of streamflow generated in the glacier-covered part of a catchment to catchment-scale water resources. It uses two contrasting glacierisedHimalayan catchments, one of which is winter-precipitation dominated, Chandra (the western Himalaya), and the other one summer-precipitation dominated, upper Dudhkoshi (the eastern Himalaya). For these catchments, climate sensitivities of simulated streamflow is obtained by regressing the simulated variability of streamflow to the one its meteorological drivers. The used model is a the Variable Infiltration Capacity (VIC) model, augmented with a glacier melt module.

The analysis is model-based; , the used precipitation-glacier-melt-streamflow model is very simple for the glacier-covered catchment part; as far as I see, it sums up the ice melt and the snowmelt (and rainfall) and routes it through a single (or perhaps two, unclear) linear reservoir, i.e. the corresponding streamflow response has a single time scale stemming from icemelt and snowmelt and no baseflow, thus the model can most likely not simulate a water carry-over effect from month to month for the glacier part. This model structure might have a different impact on the estimated sensitivities for the different analysed catchments.

We beg to differ with the reviewer's opinion here that our glacier model is inadequate. As shown in Table R1, the present model compares favourably with those used in the existing studies in the region.

I would like to apologize for imprecise reading; the use of two linear reservoirs was indeed stated in the original paper, but I overlooked it. This could be made clearer in the model scheme. Besides: are glacier melt, snowmelt and rainfall summed up before entering the two linear models or are they routed separately? The reference probably says it but would be good to have here)
Thank you for pointing out this ambiguity in our statement. We have modified it as (L163): "*The snow melt, ice melt, and rainfall on the glaciers routed using a linear reservoir model (Hannah and Gurnell,2001), obtained the glacier runoff. The model used two parallel reservoirs: a slow reservoir with time constant $K_{slow}$ for routing the snowmelt, and a fast reservoir with time constant $K_{fast}$ for routing the sum of the icemelt and the rainfall (Hannah and Gurnell, 2001).*"

Only two parameters of the hydrological model are calibrated, the ones that affect the water balance the most strongly (melt factor for ice and precipitation scaling factor). The calibration is on streamflow and glacier mass balance; there is an empirical weight factor to combine the performance with respect to both quantities; despite a clear lack of giving any formal statistical framework, the parameter estimation approach is called a Bayesian Inference.

High Himalayan catchments, like the one studied here, are data-sparse. So we use a minimal set of two calibration parameters to avoid over-fitting (L 186). All the other parameters were assigned reasonable values, and the corresponding sensitivity was shown to be small (L 352–358). We believe calibrating a small subset of the model parameters is a common compromise in hydrological modelling (eg, Table R1).

We modified the conditional probability (Eq 2) by including both the model and observation errors (Eq 2) and by removing the empirical weight of 1/2. Please refer to the revised Sect. 3.2.3 for the details. The present approach of using uniform prior distributions for the two fit-parameters (L 191), using a Bi-variate Gaussian distribution for the residues associated with the two (independent) observed datasets (L 206), and obtaining a posterior probability distribution for the models (shown explicitly in Fig 4a, 4b), is a Bayesian approach to the best of our understanding.

Thanks for having clarified and corrected the conditional probability formulation, accounting now for the model residuals (instead of simply the observational error) and removing the factor ½. Because of the omission of the model residuals and the empirical factor, the chosen approach was unclear and appeared to me an ad-hoc calibration criterion; it is clearer now. As far as I see, it is not mentioned in the revised version that you make an explicit assumption about the distribution of the model residuals (only in the response), which might be good for non-expert readers. It is indeed common to make the normality assumption for discharge residuals; I am not sure if there are any examples of assumption about mass balance residuals (this is a detail of course).

Thank you for pointing out this critical limitation of our formulation in your earlier comments, which helped improve our understanding. We have now explicitly stated the assumption in the main text L206: "*The conditional probability p(d|θ) of the observations d given the model parameter θ was assumed to be a bivariate normal distribution (e.g., Rounce et al., 2020; Werder et al., 2020), i.e., a normally distributed residuals for both discharge and glacier mass balance...*"

We confirm that both of the cited references, Rounce et al., 2020 and Werder et al., 2020, assumed normally distributed mass-balance residuals.

Furthermore, I now understand that you sampled the full 2D parameter space but kept the corresponding residual variance constant to a fixed value to compute the full posterior probability. In my experience, the residual variance is often sampled along with the parameters. The chosen solution is pragmatic (especially given the few mass balance observations) but not the standard procedure.

We now make this point explicit L215: "*…incorporated the errors in the model ($\sigma^{mod}$) and the observation ($\sigma^{obs}$). Each of these errors was assumed to be a constant having the following values. $\sigma_Q^{obs}$ was…*"

moreover, the conclusion is very general with new insights that can be inferred from general process knowledge such as e.g. the sentence "the temperature sensitivity of the glacier runoff and the precipitation sensitivity of the off-glacier runoff are critical determinants of the future changes of summer runoff and its variability in these two catchments". I therefore recommend rejection of this version.

Again, we beg to differ as,

● We are not aware of any study in the Himalaya where the above pattern of sensitivity has been quantified using glaciohydrological modeling and explained in terms of the underlying processes.
● We are not aware of any studies where the implication of the above property on the variability and change in runoff of any Himalayan catchment have been analysed.

I still maintain my comment. The take home messages of the analysis read as follows (full relevant text of the paper copied below):

"The precipitation sensitivities of the summer runoff of the non-glacierised parts of the catchments are high, but those of the glacierised parts are negligible. In contrast, the temperature sensitivities of summer runoff of glaciers are high, but those of the non-glacierised parts are negligible. As a consequence, the temperature sensitivity of the glacier runoff and the precipitation sensitivity of the off-glacier runoff are critical determinants of the future changes of summer runoff and its variability in these two catchments. "

We could have guessed that glacier runoff is temperature sensitive (in a model that uses only temperature as driver for melt) and that the off-glacier parts are sensitive to precipitation input. The abstract gives some more interesting text but the conclusion is definitively not yet there.

We re-written the 'Summary and conclusions' section' with additional statements related to the implications of the observed pattern of sensitivities on runoff variability.

Why does the glacier melt model not use the energy-balance approach?

Due to the scarcity of field data, we chose a minimal DDF model for the ice melt, following most of the hydrological studies in the region (see Table R1).

Ok, but the snowmelt model of VIC already uses an energy balance approach, i.e. the energy balance approach is set-up for snow melt, this means that all data is available? Or do I get this wrong? Accordingly, my question still holds: why is the energy balance approach feasible for the snow on the glacier but not for the ice melt under the snow? If this is simply for practical reasons (coding), it is perhaps worth mentioning.

We now clarify that (L159): "*A minimal temperature-index model (Hock, 2003) was chosen to simulate the ice melt over the corresponding snow-free areas. This one-parameter model is easy to calibrate and is expected to work well for ice cover than for snow cover, due to a relatively low seasonal variability of ice albedo (Hock, 2003).*"

Is the glacier melt coded by the authors of the study or someone else?

We do not understand the relevance of the question.

We have given due credit to any piece of code used in this paper that is not written by us.

I asked the question because it is unclear if the code is available, which is always interesting for other users.

The developed glacier model and glacier routing model are available in this public domain: https://osf.io/7une2/. All the observed hydrometeorological will be made public after publication.

Methods: the computation of glacier mass balance sensitivity is not clear to me; did you run the model with modified precipitation input?

As already stated in L 270, we did not perturb the precipitation but used the interannual variability of mass balance to compute the sensitivities (just as we do for the runoff), L 270: "Apart from the sensitivities of summer runoff, we also computed the precipitation and temperature sensitivities of glacier mass balance using the corresponding simulated interannual variability over the period of 1980–2018.

I still do not understand. Please not that my question arose due to the sentence following the one above.

We apologise for the unclear statement, which is modified to L273: "*On the glacierised part, we estimated the mass-balance sensitivities to the corresponding temperature and precipitation forcing over the period of 1980–2018. The sensitivities were computed by linearly regressing the modelled anomalies of the mass-balance to those of the annual precipitation and summer air-temperature.*"

how high is ET?

ET is about ~⅓ of the annual precipitation in these catchments (Fig S7).

Ok, is this in the main text now, I think this is highly relevant.

We thank you for raising the point.